# REFACTORING CODEBASES THROUGH LIBRARY DESIGN

## ABSTRACT

Maintainable and general software allows developers to build robust applications efficiently, yet achieving these qualities often requires refactoring specialized solutions into reusable components. This challenge becomes particularly relevant as code agents become used to solve isolated one-off programming problems. We investigate code agents' capacity to refactor code in ways that support growth and reusability. We first investigate what makes a good refactoring, finding via asymptotics analysis and a human study that Minimum Description Length best aligns with developer preferences for code refactoring quality. We then present both a benchmark and a method for refactoring: MINICODE, a benchmark where multiple files must be refactored into a shared library, and LIBRARIAN, a sample-and-rerank method for generating reusable libraries. We compare LIBRARIAN to state-of-the-art library generation methods, and study it on real-world code bases.

## 1 INTRODUCTION

Writing code is mainly a matter of *re*writing code: debugging, refactoring, optimizing, and other activities within the software engineering lifecycle. But poor rewrites incur technical debt, with such debt costing up to $2 *trillion* annually (Tews). This problem will likely worsen as language models become increasingly responsible for generating code, because they excel at solving isolated programming problems, but their context length demands a myopic view of the codebase. It is therefore valuable to understand not just the ability of language models to solve programming problems, but also their ability to rewrite and refactor code in ways that support growth and reuse.

Effective code refactoring at scale is a design problem whose concerns center on re-usability and maintainability. A classic example illustrates this design challenge: Human programmers often create overly-specialized, redundant solutions to similar problems and would benefit from redesigning specialized solutions into a shared library. This consolidation requires careful design decisions about the right level of abstraction — neither too specific nor too general — and appropriate interfaces that balance flexibility with usability.

Here we focus on refactoring multiple files into a reusable software library, which raises two questions: (1) How should we quantify the quality of a library refactoring, and (2) To what extent can LLMs generalize code into reusable libraries? To answer those question, we develop a new method and a benchmark. This goes beyond past work (Wong et al., 2021; Stengel-Eskin et al., 2024; Ellis et al., 2021; Bowers et al., 2023; Dechter et al., 2013; Grand et al., 2023; Liang et al., 2010b) in *library learning* that synthesized subroutines across small programs in i.e. $\lambda$-calculus, instead tackling the more naturalistic problem of redesigning code written in contemporary high-level languages, such as Python, producing classes, methods, and helper functions in the style of a human-written library. We develop a method, LIBRARIAN (Figure 1), which samples possible code rewrites and then reranks those samples based on criteria designed to capture a good refactoring. To generate potential rewrites, we develop methods for clustering pieces of code together that share common structure so that a succinct prompt can rewrite them jointly into their refactored form. To find strong criteria for ranking potential rewrites, we study a variety of metrics across machine learning and software engineering, both on programming benchmarks and via a human study.

To evaluate our method and systematically assess the capability of current agents to generate libraries, we introduce a new benchmark, MINICODE, which addresses three key desiderata missing from existing benchmarks. First, open-ended design: unlike SWE-Bench (Jimenez et al., 2024), Commit0

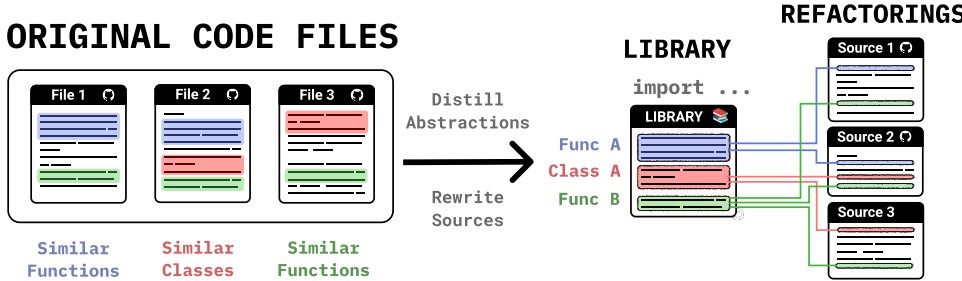

Figure 1: Overview of the refactoring problem. A refactoring task comprises a set of files. We refactor the files by designing a new library. Candidate refactorings are evaluated based on a refactoring metric, and are expected to maintain correctness of the original code sources (pass rate). We explore several refactoring metrics in this paper.

(Zhao et al., 2025), and RefactorBench (Gautam et al., 2025) which primarily focus on functional correctness, MINICODE presents an unconstrained library design problem. Agents create a library that can be imported back into a repository, with complete freedom to design the interface and implementation from scratch—optimizing for software engineering objectives like reusability and maintainability. Second, verifiability: we ensure objective evaluation by retaining the unit tests from all repositories that will import the designed library, allowing us to verify that the solutions work correctly across multiple use cases. Third, large context: agents must understand and synthesize information from multiple code sources (files) simultaneously to design a unified library that consolidates specialized code sources into a general interface.

We contribute the following:

1. Study of different metrics for library creation via a user study and program synthesis benchmarks, with the surprising finding that neural language models offer a stronger signal of refactoring quality than classic metrics from the software engineering community.

2. A new benchmark, MINICODE, covering both competition programming and real-world machine learning programs from the Transformers and Diffusers libraries.

3. A new algorithm, LIBRARIAN, which outperforms prior library learners on existing benchmarks and scales to complex codebases.

4. Refactorings of real-world codebases widely used in the machine learning community. Our method successfully refactors the Transformers and Diffusers libraries from Huggingface to be shorter and more reusable. To the best of our knowledge this is the first time library learning has have been successfully applied to real-world software projects.

## 2 RELATED WORK

**Library Learning.** Systems which perform library learning research discover shared abstractions across a large number of small programs, which they use to automatically define new subroutines. Systems such as DreamCoder (Ellis et al., 2021), Trove (Wang et al., 2024), LiLo (Grand et al., 2023), and REGAL (Stengel-Eskin et al., 2024) automatically construct such libraries with the goal of making future program synthesis tasks easier to solve, once the learned library is in hand. Our work is closest to REGAL (Stengel-Eskin et al., 2024), which clusters related code and refactors using language models. However, existing library learning approaches have primarily been demonstrated in small-scale, constrained domains, limiting their applicability to typical software engineering tasks, such as consolidating multiple repositories into cohesive libraries. By framing library learning within the context of realistic, large-scale code repository development, we expand the relevance of library learning to everyday software engineering practice.

**Repo-level coding benchmarks.** Recent work has explored the application of language models to repository-level software engineering tasks. Existing benchmarks include SWE-bench (Jimenez et al., 2024), which evaluates models on their ability to resolve real-life GitHub issues, and Commit-0 (Zhao

et al., 2025), which requires agents to fill in function definitions. Such benchmarks primarily evaluate functional correctness via unit tests, without assessing the quality or maintainability of the resulting codebase. RefactorBench (Gautam et al., 2025) takes a step in this direction by benchmarking the ability to follow specific refactoring instructions. Our work differs by requiring models to perform a more open-ended task: Redesigning code to be more modular and compact by discovering reused abstractions, while retaining verifiability by re-using downstream unit tests. Additionally, libraries must be created without any scaffolding limitations such as preexisting function definitions affording more design freedom than Commit-0.

**Program optimization.**   While our goal is to optimize the quality of libraries, other works focus on improving execution speed through correctness-preserving transformations (Waghjale et al., 2024; Ouyang et al., 2025; Schkufza et al., 2013). Both forms of program optimization, compression and speed, are more open-ended than optimizing only for correctness, as there does not exist a ground-truth answer. Prior work on program optimization benchmarks study code at the file level. We propose a benchmark that transforms programs at a larger scale, across multiple code files.

## 3 PROBLEM STATEMENT

Given $N$ related files $\{\rho_n\}_{n=1}^N$, the goal is to create a library $\mathcal{L}$ that captures shared abstractions. The original files $\{\rho_n\}_{n=1}^N$, which define the refactoring problem, we call a **task**. The new library must support all original use cases in the task by extracting latent shared abstractions. This is accomplished by searching for refactorings that are both correct and 'natural'. Correctness is straightforward to define via unit tests, but naturalness is more challenging to quantify.

Shorter code is potentially simpler and less redundant. One potential metric is to count the number of tokens, lines-of-code, or syntax tree nodes in the proposed library and refactored code (Dechter et al., 2013; Polozov & Gulwani, 2015; Bowers et al., 2023; Cao et al., 2023). But minimizing program size has obvious failure modes: code should also be understandable and extensible, which can be in tension with merely finding the shortest program.[1] Other work in program synthesis (Liang et al., 2010a; Solomonoff, 1964; Ellis et al., 2021) instead optimizes *Minimum Description Length* (MDL), or negative log probability under a reference distribution. In the software engineering community, other metrics such as cyclomatic complexity and maintainability index have been defined for similar purposes: These are more complex metrics that examine the syntax tree, call graph, and other statically-analyzable structures (McCabe, 1976). What metric should we use? We revisit this question in Section 6, where we empirically compare candidate metrics and human preferences before fixing our choice for the rest of the paper.

For now assume a placeholder metric $M$ measuring refactoring quality; we seek to minimize $M$ while preserving correctness. Given a task comprising files $\{\rho_n\}_{n=1}^N$, we output both a new library $\mathcal{L}$, as well as rewritten refactorings of the original files, $\{\rho'_n\}_{n=1}^N$. We define the pass rate $\tau(\rho_n)$ as the set of unit tests $\rho_n$ passes, and consider both refactoring several files ($N > 1$) and also refactoring a single large file ($N = 1$). We optimize the following objective, which prefers passing at least the same tests as the original programs *and* minimizing the chosen metric $M$:

$$\ell\left(\mathcal{L}, \{\rho'_n\}\right) = \begin{cases} M(\mathcal{L}, \{\rho'_n\}) & \forall \rho_n, \tau(\rho_n) \subseteq \tau(\rho'_n) \\ \infty & \text{otherwise} \end{cases} \tag{1}$$

## 4 MINICODE—LIBRARY DESIGN AND REFACTORING BENCHMARK

MINICODE presents systems with a task comprising a set of files, then asks them to refactor the files into a unified library alongside refactorings of the original files. There are two key desiderata for benchmark tasks: They should have related files sharing latent abstractions, and should also be verifiable, to measure how well refactored files preserve functional correctness. We source a variety of problems (Table 1).

---

[1]Perl Golf is a game where participants attempt to write the shortest Perl program accomplishing a given task. The resulting code is famously incomprehensible, even by the standards of Perl.

Table 1: MINICODE Statisics

| Domain | Files | Tasks | Avg LoC | Avg Tests / file |
|--------|-------|-------|---------|------------------|
| Code Contests (Li et al., 2022b) | 300 | 10 | 87 | 10 |
| Transformers (Wolf et al., 2020) | 10 | 1 | 538 | 181 |
| Diffusers (von Platen et al., 2022) | 11 | 2 | 685 | 75 |
| Logo (Wong et al., 2021) | 300 | 1 | 10 | 1 |
| Date (Srivastava et al., 2023) | 246 | 1 | 14 | 1 |

**CodeContests.** Competition problems are crafted with specific variations of algorithmic approaches in mind, resulting in both shared latent concepts and the required test cases. As a result, competition coding is both verifiable, and ready to refactor. We therefore take solutions, prompts, and tests from CODECONTESTS (Li et al., 2022a), a competition programming dataset.

**Huggingface Transformers Library.** We test refactoring across implementations of large language and vision–language models from the Huggingface `transformers` repository (`modelling_<name>.py` files, e.g., Qwen2, LLaMA, DeepSeek-V3). Unlike competition coding, these sources are production-scale and Huggingface requires that all changes pass an extensive suite of integration tests before merging into the main branch. A refactoring is only deemed correct if it passes the unmodified Transformers test suite, making this a high-stakes setting that requires correctness and compatibility.

**Huggingface Diffusers Library.** We test refactoring across implementations of diffusion models from the Huggingface `diffusers` repository (`unet_<name>.py` and `scheduler_<name>.py` files, e.g., Stable Diffusion UNet, DDPMScheduler), yielding two distinct tasks. Like Transformers, Diffusers requires that all changes pass a comprehensive suite of integration tests before merging into the main branch.

**Logo & Date.** The library learning literature already has existing benchmarks: Typically they seek to learn a single library from a task comprising many sources, and then test that library on holdout program synthesis tasks. Logo and Date were used in the recent related work REGAL (Stengel-Eskin et al., 2024), which we incorporate wholesale to understand how our new method compares to state-of-the-art library learning. The associated programming problems were created by humans, but their solutions were generated by gpt-3.5-turbo.

## 5 LIBRARIAN: REFACTORING CODE TO CREATE LIBRARIES

LIBRARIAN generates a new library from a set of files, while migrating the files to use that new library (Figure 1), following a sample-and-rerank framework: Prompting a backend LLM or agent to sample $K$ candidates, and picking the one minimizing the loss $\ell$. Naively,

$$\mathcal{L}, \{\rho'_n\} = \underset{\mathcal{L}, \{\rho'_n\} \in \text{SAMPLE}_K(\{\rho_n\})}{\arg\min} \ell\left(\mathcal{L}, \{\rho'_n\}\right) \tag{2}$$

for metric $M$ and sampling budget $K$. But this cannot work for large tasks with many programs, which would not fit into the context of most LLMs. Even long context models cannot process the entirety of e.g the Linux kernel, and even if they could, it is not clear that such a strategy is the most efficient way of focusing the language model's attention. To address this, we wrap sample-and-rerank with a clustering algorithm that decomposes the task into manageable chunks, described next.

**Clustering.** Meaningful abstractions arise when programs share underlying functionality or structure. To surface these, we cluster the task's files into small groups that are likely to share reusable structure, and refactor each cluster separately from the rest. This decomposition shrinks the prompt size, and gives independent searches for the best per-cluster refactoring, which may be more tractable.

LIBRARIAN's clustering extends REGAL (Stengel-Eskin et al., 2024), which clusters programs by assuming each program is paired with a natural language description of the problem it solves, and clustering embeddings of those descriptions. Since similar problems need not imply similar

solution code, we instead prompt a model to summarize each file and cluster by these summaries. Specifically, we define $\text{CLUSTER}_S(\{\rho_n\})$ as performing agglomerative clustering (Ward Jr, 1963) on the task's files $\{\rho_n\}$ to produce a set of set of files, each of which is a cluster of size $S$. We use text-embedding-ada-002 to embed descriptions of code sources for clustering.

**Combining clustering with sample-and-rerank.** The simplest approach is to refactor each cluster independently and take the union (concatenation) of each cluster's library:

$$\mathcal{L}^\star = \bigcup_{c \in \text{CLUSTER}_S(\{\rho_n\})} \mathcal{L}_c \tag{3}$$

$$\{\rho'_n\} = \bigcup_{c \in \text{CLUSTER}_S(\{\rho_n\})} \{\rho'_{c,i}\}_{i=1}^{|c|} \tag{4}$$

$$\mathcal{L}_c, \{\rho'_{c,i}\}_{i=1}^{|c|} = \underset{\mathcal{L}, \{\rho'_{c,i}\}_{i=1}^{|c|} \in \text{SAMPLE}_K(c)}{\arg\min} \ell\left(\mathcal{L}, \{\rho'_{c,i}\}_{i=1}^{|c|}\right), \text{for each } c \in \text{CLUSTER}_S(\{\rho_n\}) \tag{5}$$

The approach above ignores the fact that library abstractions discovered in one cluster might be useful in another cluster. A more sophisticated approach accumulates a library across clusters, and when refactoring a cluster, adds the accumulated library to the prompt. This lets abstractions discovered earlier carry forward across the collection. Appendix A describes this extension.

# 6 WHAT MAKES A GOOD REFACTORING?

We compare different metrics $M$ measuring the quality of a refactoring:

**Tokens** measures the total number of tokens in the refactored files *and* in the library. It minimizes program size, but not at the expense of creating a bloated library: Simply replacing every program with a its own monolithic library function would not improve the tokens metric, because it measures library size as well. Concretely, $M_{\text{tokens}}(\mathcal{L}, \{\rho'_n\}) = \text{TOKENS}(\mathcal{L}) + \sum_n \text{TOKENS}(\rho'_n)$.

**Minimum Description Length (MDL)** evaluates the negative log probability under a reference distribution, taking into account both the library and refactored sources. Concretely, $M_{\text{MDL}}(\mathcal{L}, \{\rho'_n\}) = -\log p_{\text{LM}}(\mathcal{L}) + \sum_n -\log p_{\text{LM}}(\rho'_n \mid \mathcal{L})$, where $p_{LM}(\rho'_n|\mathcal{L})$ is concatenating the library and the program into one prompt, but only counting the perplexity of the later program tokens. This has a Bayesian justification: The MDL library is the maximum aposteriori estimate of $\mathcal{L}$ given conditionally-independent code sources. We use Qwen-2.5-3B as our reference language model, as it is modern, open, and has publicly-available endpoints for querying logits, which is required for scoring refactorings. To confirm that our MDL optimization results are not model specific, we computed MDL values for 15 Code Contests clusters, each of 50 valid refactoring candidates, using Qwen and Llama-3.2-3B models. We found a 94% agreement in the minimum MDL candidates from both models.

**Cyclomatic Complexity (CC)** is a longstanding metric from the software engineering community which measures the number of linearly independent paths through a program's control flow graph (McCabe, 1976). Smaller programs often have lower cyclomatic complexity. It is equivalent to defining $M_{\text{CC}}(\mathcal{L}, \{\rho'_n\}) = \text{CC}(\mathcal{L}) + \sum_n \text{CC}(\rho'_n)$ and $\text{CC}(\rho) = E(\rho) - N(\rho) + 2P(\rho)$, where $E$, $N$, and $P$ measure the number of control flow edges, nodes, and connected components, respectively.

**Maintainability Index (MI)** is a modern software engineering metric combining several other metrics, including lines-of-code, cyclomatic complexity, and Halstead volume into a single score. Higher MI values are intended to indicate easier-to-maintain code, so we define $M_{\text{MI}}(\mathcal{L}, \{\rho'_n\}) = -\text{MI}(\mathcal{L}) + \sum_n -\text{MI}(\rho'_n)$.

## 6.1 ASYMPTOTIC BEHAVIOR OF METRICS IN LARGE-SAMPLE REGIME

Are these metrics equally effective at encouraging modular and reusable libraries? To answer this question, we run LIBRARIAN on 15 CodeContests (each of three files) using MDL, tokens, maintainability index, and cyclomatic complexity, while varying the inference-time sample budget $K$ (Figure 2). We use Best@k estimator for the expected value of metrics for all $k \leq K$ (we describe the estimator and prove its correctness in Appendix F ). Tokens and MDL separate cleanly from classic software engineering metrics: Optimizing tokens/MDL, both of which essentially compress

the original programs, does *not* yield steady improvements in MI/CC, and vice-versa. To understand whether these libraries expose shared abstractions, we examine the average number of times that each library routine is used, and the average number of library invocations per library function. This teases apart tokens and MDL: Optimizing MDL yields more more reusable libraries (used about $8\times$ per task), with each function called more often (called about $2.2\times$ per function)—exceeding the other metrics we consider. See also Appendix Figure 7 for the raw libary metrics and CC results.

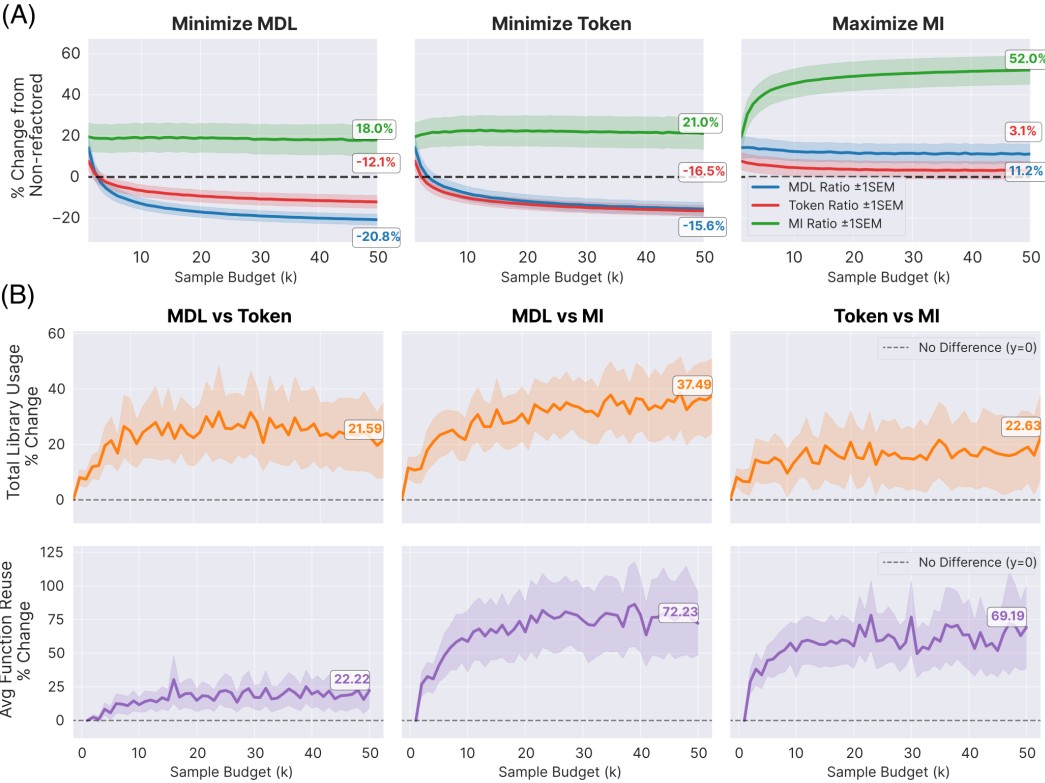

Figure 2: (A) Asymptotic behavior of metrics for scoring libraries and refactorings (columns) varying refactoring budget (horizontal axes). (B) Comparing metrics via proxies of downstream library quality (total library usage and average calls per library function), for which MDL>Tokens>MI. All results are estimated using Best@k. See also Appendix Figure 7.

Studying these metrics at large $k$ allows understanding their inference-time scaling behavior. While the underlying metric itself improves (with diminishing returns), this was not to be taken for granted: The backend language model must produce sufficient diversity to steadily improve these metrics. Prior state-of-the-art, such as Stengel-Eskin et al. (2024), instead take a single sample, but the results here suggest benefit from further test-time search, and indeed, real-world repos benefit from steady improvement with increased samples (Figure 3). But our proxies of library utility plateau much earlier, around $k = 20$ samples, suggesting large $k$ is unnecessary in practice: Effective library building benefits from test-time compute, but does not demand an exorbitant amount of it.

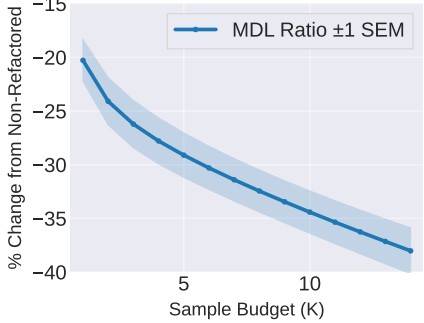

Figure 3: Best@K MDL ratio. Increasing sample budget improves MDL on Transformers.

## 6.2 HUMAN EVALUATION OF REFACTORING METRICS

Given the strong separation between compression metrics (MDL/tokens) and software engineering metrics (MI/CC), we perform a human study to corroborate the findings of Section 6.1 using the exact same CodeClusters clusters. Our human study compares tokens, MDL, and Maintainability Index by (1) refactoring CodeContest tasks into libraries, (2) presenting human participants with the original sources and their refactorings under pairs of metrics, and (3) eliciting pairwise preferences from human participants.

Humans strongly prefer the compression-based metrics (MDL/tokens) and disprefer the metrics developed within the software engineering community, but show no statistically significant difference between MDL and tokens given only $N = 12$ human participants (Figure 4). Although MDL and tokens measure different things, they often—but not always—prefer similar libraries, making it challenging to achieve the statistical power needed to tease them apart with a reasonable number of human subjects. Figure 5 illustrates the kinds of corner-cases where MDL and tokens disagree: Although such cases are uncommon, we believe basically every human coder would prefer the MDL-minimizing program.

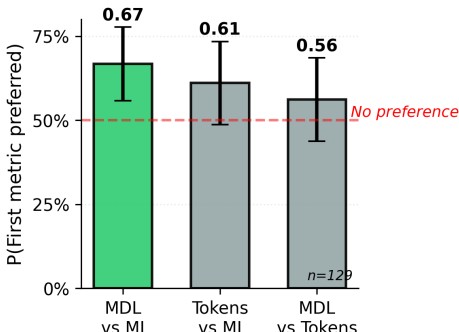

Figure 4: Human evaluation of different refactoring objectives. Judges compare pairs of refactorings that both pass all test cases. MDL aligns best with human preferences.

**We therefore adopt $M_{MDL}$ as the primary objective in the remainder of this paper:** Our human study lacked enough participants to separate tokens from MDL, but (1) Bayesian arguments support MDL; (2) corner cases in the style of 'Perl golf' provide existence proofs of the liability of merely minimizing tokens; and (3) reasonable proxies for library reuse favor MDL (Section 6.1).

```python
from ..shared_library import (
    rotate_half,
    apply_rotary_pos_emb,
    repeat_kv,
    eager_attention_forward,
    RMSNorm,
    BaseMLP,
    BaseRotaryEmbedding,
    BaseAttention,
    BaseDecoderLayer,
)

class LlamaRMSNorm(RMSNorm):
    ...

class LlamaRotaryEmbedding(BaseRotaryEmbedding):
    ...

class LlamaMLP(BaseMLP):
    def __init__(self, config):
        super().__init__(
            config,
            mlp_bias=config.mlp_bias
        )

class LlamaAttention(BaseAttention):
    def __init__(self, config: LlamaConfig,
                 layer_idx: int):
        super().__init__(
            config=config,
            layer_idx=layer_idx,
            attn_bias=config.attention_bias,
            sliding_window=None
        )
...
```

```python
from ..shared_library import rotate_half,
    apply_rotary_pos_emb,repeat_kv,
    eager_attention_forward,RMSNorm as R,
    BaseMLP as M,BaseRotaryEmbedding as E,
    BaseAttention as A,BaseDecoderLayer as
    Y
class Z(R):...
class I(E):...
class J(M):
 def __init__(s,g):
  super().__init__(g,mlp_bias=g.mlp_bias)
class H(A):
 def __init__(s,g:F,i:int):
  super().__init__(
   config=g,layer_idx=i,
   attn_bias=g.attention_bias,
   sliding_window=None)
class V(Y):
 def __init__(s,g:F,i:int):
  super().__init__(
   config=g,layer_idx=i,
   norm_class=Z,
   mlp_class=J,
   attention_class=H)
...
```

Figure 5: Example where tokens and MDL diverge: Obfuscating the original library definitions (left) by shortening variable names (right) reduces tokens but increases MDL.

## 7 WHAT WE LEARN FROM RUNNING LIBRARIAN ON MINICODE

We empirically study LIBRARIAN on MINICODE with the goal of understanding (1) the degree to which library abstractions are reused across programs, (2) how our method compares to state-of-the-art library learning on existing datasets, and (3) whether LIBRARIAN holds value for real-world repos.

**LIBRARIAN discovers reusable functions for competition programming–but some functions are only called once.** We test on CodeContests with a cluster size of $S = 3$ and a sample budget of $K = 8$ draws from o4-mini, as reasoning models perform well on competition programming.[2] Table 2 shows that the resulting refactors and libraries approximately halve the MDL, which incidentally reduces program size as well (44% relative reduction in token count). Pass rate modestly improves as an incidental consequence of sampling and filtering with test cases. Libraries average 10 functions, each heavily reused:

Table 2: Results for LIBRARIAN ($K = 8, S = 3$) on 10 Code Contests tasks.

| Metric | Value |
|---|---|
| Pass Rate | 90.67% ±1.88 |
| Pass Rate Improvement | 6.33% ±1.41 |
| MDL Ratio | 0.53 ±0.03 |
| Token Ratio | 0.66 ±0.04 |
| Library Functions | 10.30 ±1.41 |
| Avg Calls per Function | 5.17 ±1.08 |
| % Single Use Functions | 38.03% ±4.88 |

Averaging 5 uses per function within tasks comprising only 10 programs. But almost 40% of library functions are only used once. Why is that?

A signature of the MDL objective is a preference for whatever a language model assigns high apriori probability to. Although a single-use function does not reduce line count or tokens —the function could simply be inlined—it improves MDL if it yields a more natural decomposition of the target programs. Indeed, human-written libraries sometimes include functions that are seldom used, provided they serve as a conceptually modular abstraction. We therefore see single-use functions as a feature, not a bug. See Appendix J for an example refactoring candidate on CodeContests.

**Are these libraries useful for solving new, unseen programming problems?** For more than a decade library learning has sought to learn libraries from training programs which then help solve new unseen program synthesis tasks. The Logo and Date datasets fit within this paradigm. Recently REGAL improved the state-of-the-art on these library learning datasets. Because our clustering is heavily inspired by REGAL, for fair comparison, we keep exactly their clustering setup but add MDL-based reranking using $K = 5$ samples. Despite the simplicity of these

Table 3: Solving holdout test program synthesis tasks using learned libraries

| Dataset | Model | Pass Rate |
|---|---|---|
| Logo | REGAL (gpt-3.5-turbo) | 49.3% ±1.1 |
| | LIBRARIAN (3.5-turbo) | 69.9% ±0.9 |
| Date | REGAL (gpt-3.5-turbo) | 90.2% ±0.5 |
| | LIBRARIAN (3.5-turbo) | 94.7% ±0.7 |

datasets, we find value in our more complicated method. Table 3 shows that sampling and reranking by MDL yields up to a 41.8% relative improvement in solve rate on unseen programming problems, and that even when the gains are more modest, we still improve upon the state-of-the-art. But these are relatively simple problems solvable with about ten lines of code—does this work in the real world?

**Real-World Refactoring.** The HuggingFace Transformers library is used by nearly 400k GitHub projects. We deploy LIBRARIAN to 10 source files, using Claude Code to sample $K = 15$ refactorings per cluster of size $S = 5$, believing an agent such as Claude Code would excel at repo-level edits. LIBRARIAN distilled repeated abstractions such as MLPs, Attention, Decoder classes, RoPE helper functions, etc., achieving an MDL 67% of its original value while still passing all integration tests. The top-3 refactorings based on MDL have an average of $18 \pm 4.4$ abstractions (functions, classes) in the library, each of which is called on average $4.59 \pm 0.39$ times in the refactored models. For Diffusers, scheduler clusters yielded top-3 MDL refactorings with an average of $12.3 \pm 1.6$ functions and $3.0 \pm 0.4$ calls per function, while UNet refactorings produced richer abstractions with an average of $17.0 \pm 5.6$ functions/classes and $3.43 \pm 0.67$ calls each. Refactoring at scale proved expensive: Each refactoring took approximately 30 minutes to generate and test. But this is a one-off cost, and

---

[2]Code agents such as Codex, Claude Code, and others underperformed o4-mini (Appendix I)

in our view, the refactored Transformers and Diffusers sources are much cleaner, and the new library is transparently reusable (Figure 6). To the best of our knowledge, this is the first time any library learning algorithm has been successfully applied to real-world software projects.

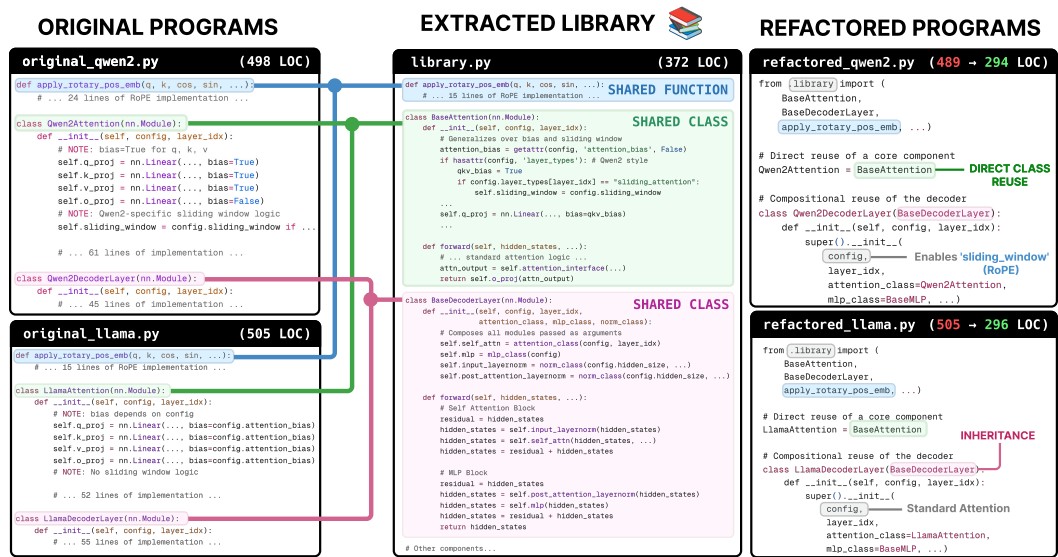

Figure 6: Representative result for refactoring HuggingFace Transformers using LIBRARIAN

**Learned libraries from these real-world codebases are useful for unseen downstream refactoring tasks.** When a library learned on one cluster of Transformer files (5 models) is applied to refactor a second cluster, LIBRARIAN reduces the unseen cluster's MDL to 73% of the its original value, with an average of 3.0 calls per library function. This demonstrates that LIBRARIAN learned libraries that can be repurposed to more compactly rewrite unseen real-world code sources.

## 8 CONCLUSION

We introduce a new benchmark MINICODE and method LIBRARIAN for compressing files through reusable abstractions. We highlight the challenges of producing modular and maintainable libraries, then present an effective method for using LLMs to do this task. By framing refactoring as an optimization problem, our work opens new directions for building more general and scalable code understanding and generation systems. In particular, the structure of MINICODE lends itself well to reinforcement learning, where training would entail synthesizing collections of repositories to refactor then computing rewards based on MDL or other metrics.

**Limitations.** We evaluate on synthetic toy problems (Logo and Date), and on competition programming problems, neither of which are naturalistic, although this is partly counterbalanced by our study of real-world refactoring. Compression, whether measured by tokens or MDL, may not always correlate with reuse, a limitation we sought to address through our experiments on downstream programming problems, and on holdout Transformers files: But investigating reuse on unseen app-building problems for real-world repo-level refactors remains open.

## ACKNOWLEDGEMENTS

LLMs were used to polish phrasing of some parts of the paper.

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
