# A  INCREMENTAL VERSION OF LIBRARIAN

We support sharing library components across clusters through the below incremental version of our algorithm. It processes clusters sequentially, rather than in parallel, and its output can depend on the ordering imposed upon the clusters.

$$\mathcal{L}_0 = \varnothing$$

$$\{\rho'_n\} = \bigcup_{c \in \text{CLUSTER}(\{\rho_n\})} \{\rho'_{c,i}\}_{i=1}^{|c|}$$

$$(\Delta\mathcal{L}_c, \{\rho'_{c,i}\}_{i=1}^{|c|}) = \arg\min_{(\Delta\mathcal{L}, \{\rho'_i\}) \in \text{SAMPLE}_k(c; \mathcal{L}_{t-1})} \ell(\mathcal{L}_{t-1} \cup \Delta\mathcal{L}, \{\rho'_i\}_{i=1}^{|c|}) \quad \forall c \in \text{CLUSTER}(\{\rho_n\})$$

$$\mathcal{L}_t = \mathcal{L}_{t-1} \cup \Delta\mathcal{L}_t, \qquad \mathcal{L}^\star = \mathcal{L}_{|\text{CLUSTER}(\{\rho_n\})|}$$

# B  ALGORITHM

---

**Algorithm 1** Refactoring Specialized Programs into a Joint Library

---

**Require:** Set of independent, specialized programs $P_{initial} = \{\rho_1, \rho_2, \ldots, \rho_n\}$
**Require:** Sample Budget $K$
**Ensure:** Joint library $\mathcal{L}_{final}$ and set of refactored programs $P_{final}$
 1: $\mathcal{C} \leftarrow \text{Cluster}(P_{initial})$
 2: $\mathcal{L}_{final} \leftarrow \emptyset, P_{final} \leftarrow \emptyset$
 3: **for all** cluster $c \in \mathcal{C}$ **do**                                    ▷ Each cluster independently
 4:     $T_C \leftarrow \text{GroupIntoTuples}(c)$                          ▷ Get tuples for each cluster
 5:     **for all** tuples $\tau \in T_C$ **do**
 6:         $\{f_{retrieved}\} \leftarrow \text{RetrieveRelevantFromLibrary}(\mathcal{L}, \tau)$
 7:         $S \leftarrow \emptyset$
 8:         **for** $i = 1$ to $K$ **do**                                    ▷ Sample $k$ times
 9:             $(\{f_{new,i}\}, \{\rho'_i\}\}) \leftarrow \text{SAMPLE}(f_{retrieved}, \tau)$
10:             $S \leftarrow S \cup \{(\{f_{new,i}\}, \{\rho'_i\})\}$
11:         **end for**
12:         $(f_{best}, \{\rho'_{best}\}) \leftarrow \text{RerankAndSelectBest}(S, \ell(\cdot))$     ▷ Rerank using objective
13:         $\mathcal{L}_{final} \cup \{f_{best}\}$
14:         $P_{final} \cup \{\rho'_{best}\}$
15:     **end for**
16: **end for**
17: **return** $\mathcal{L}_{final}, P_{final}$

---

# C  EXPERIMENTAL SETUP

**Grouping Programs into Collections**  To facilitate parallel application of LIBRARIAN and manage the dataset scale, we assume that semantically distant files will have minimal overlap in their optimal library functions. Therefore, our overall approach partitions the dataset into disjoint collections through clustering.

For **CodeContests**, these collections are constructed from an initial corpus of ~9k problems with Python solutions: We first filter these files, removing those whose selected canonical solution is under 10 lines (minimal refactoring potential). For the remaining 4596 solutions we use a language model to generate textual descriptions of canonical solutions—emphasizing reusable components—which are embedded using OpenAI's `text-embedding-ada-002`.

Agglomerative Clustering (Ward Jr, 1963) is subsequently applied to these embeddings to partition the files into a predefined number of initial clusters, in our case 120. To create uniformly sized experimental units, we subsample each such cluster to form collections of 30 files. This collection

size was empirically chosen because it balanced between the runtime of LIBRARIAN without limiting compression. We select 10 collections that we then use to evaluate our methods.

For Transformers, since the number of models is on the lower end, we manually chose a set of popular LLM / VLM models and passed them to the agent in collections of 5 code sources.

**REGAL Baselines.** To evaluate the ability of our libraries to support reuse on new problems, we turn to the program synthesis tasks used in REGAL, where learned libraries are added to help the program synthesizer. We evaluate on the two domains published by the authors, Logo and Date. Because our clustering is inspired by REGAL but adds additional complexity, for fair comparison, we keep their setup the same and only augment the training using sample + MDL rerank procedure described in Section 5.

**Code Contests.** To evaluate LIBRARIAN on refactoring Code Contests we select 6 collections of 30 files (problems). In each collection we group the problems into tuples of size 3. We set the sample budget to be $K = 8$, since our ablations show that with larger $K$ we discover better libraries 2. We use the MDL objective for rankings. The model used for sampling is OpenAI's o4-mini (OpenAI, 2024). To obtain MDL scores we use Qwen 2.5 7B Instruct (Qwen et al., 2025) as a balance between quality, speed, and cost.

**Code Agents on Transformers and Diffusers Repositories.** To fairly evaluate performance on the task by state-of-the-art systems, we use coding agents that advertise long-context ability to reason about, write, and refactor code repositories. Specifically, we use Claude Code (Cl) (Anthropic, 2025) which uses the Opus 4.1.

We test whether code agents can refactor collections of code sources autonomously, without human intervention. Refactoring repositories with code agents involves planning and iterative (re-)implementation and testing. Code agents are prompted to perform each of these steps, with feedback from the unit tests. Agents must run and repair unit tests autonomously. We run coding agents multiple times per task, logging their progress in checklists stored in text files.

The instruction provided to human evaluators is as follows:

```
## 1. Materials Provided

You will be given a set of files for each example case:

* **`original_programs.py`**: This file contains a set of 3 distinct Python programs, each presented with its
    corresponding problem description/query. This represents the "before" state.
* **`v1.py`**: This file presents the first refactoring approach. It includes:
    * The 3 refactored versions of the original programs.
    * A "library" section (e.g., `codebank.py` or inline) containing helper functions. These helper functions
      might be retrieved from an existing common library or newly created during this refactoring.
    * Either the retrieved or the new helper function sections may be _empty_, in case no programs existed in
      the codebank at the time or if no need helper functions were created by the LLM.
* **`v2.py`**: This file presents the second, alternative refactoring approach. Similar to `refactoring_v1.py
    `, it includes:
    * The 3 refactored versions of the original programs (using a different strategy than v1).
    * A "library" section with its own set of helper functions.

**NOTE**:  both refactorings had accuracy at least as good as the original programs.

## 2. Your Task

Your primary task is to:

1.  **Review** the `original_programs.py` to understand the initial code and the problems being solved.
2.  **Analyze** both `refactoring_v1.py` and `refactoring_v2.py`. Pay close attention to how the original
      programs have been restructured and what functionalities have been extracted into their respective
      libraries.
3.  **Decide which refactoring (Version 1 or Version 2) you believe is "better,"** based on the evaluation
      criteria provided below (or your own criteria!).

## 3. Evaluation Criteria: What to Consider for Your Choice

When comparing `refactoring_v1.py` and `refactoring_v2.py`, please *consider* the following aspects to inform
      your choice. The "better" refactoring should ideally excel in these areas:

> Most importantly, make sure that the extracted functions are **actually reusable and not too specific.** If
      the main programs are short, the refactoring is not immediately "better"! Try to think whether the
      extracted functions could actually be used in a different program down the line.

* **Reusability of Helper Functions :**
    * **Generality:** Are the new helper functions general-purpose and potentially useful for *other,
      different* programs and problems beyond the three presented?
    * **Reuse:** How much were existing helper functions reused?
    * **Specificity:** Are the functions too specialized to the current set of problems, limiting their
      broader applicability? _Avoid functions that are essentially just the original program broken out into a
      "helper."_
    * Composability
* **Maintainability:**
    * Readability & Understandability
    * Ease of Modification
    * Separation of Concerns

## 4. What NOT to Focus On:

* **Comments:** Please disregard the presence or absence of comments in the code for this evaluation. These
      are superficially generated by LLMs in some occasions and could be added manually after with a single
      pass.
* **Minor Stylistic Differences:** Do not focus on trivial differences in variable naming or formatting,
      unless they significantly impact readability or understanding.

## 5. How to Provide Your Feedback

For each example case, please provide:

1.  **Your Preferred Version:** (e.g., "Version 1" or "Version 2")
```

Listing 1: Human Evaluation Instruction

# D  ASYMPTOTIC BEHAVIOR RESULTS

Here we include the full graph of asymptotic behavior of all four scoring metrics (MDL, tokens, MI and CC), reporting the raw library metrics and the metric ratios. We can see that refactored programs have higher CC than the baseline and that optimizing CC as an objective does not decrese the other metrics. MDL performs best on total raw library usage as well as on average times a library function is used in the refactorings. MI ends up optimizing the number of library functions the best, but their reusability is below the average for the refactorings. Optimizing for tokens produces smaller libraries with less usage per function compared to optimizing MDL.

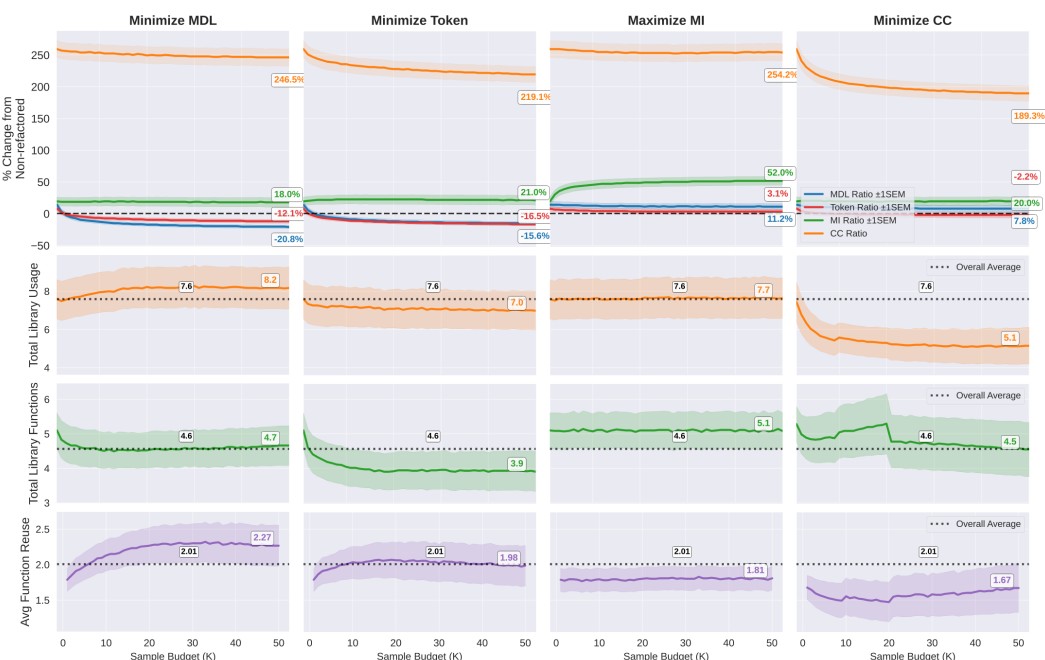

Figure 7: Asymptotic behavior of metrics for scoring libraries and refactorings (columns) varying refactoring budget (horizontal axes).

# E HUMAN STUDY DETAILS

We ran a user study with 12 participants where each participant had to judge 10 refactoring pairs. Each pair was comparing two out of three metrics, MDL, tokens, and MI. We showed the participants the original programs, as well as both of the refactoring versions (including the programs and the learned libraries). The participants were able to choose either version or say that the two refactorings are almost the same.

To quantify pairwise preferences between refactoring metrics, we employed the Bradley-Terry model, a standard framework for analyzing paired comparison data. We fit the model using maximum likelihood estimation with mutual information (MI) as the reference category ($\pi_M I = 1$). To address potential noise in human judgments, we applied consensus-based filtering with a 75% threshold, retaining all responses for comparisons with low consensus (indicating genuine ambiguity) while excluding minority responses on high-consensus comparisons where the majority preference likely indicates the correct judgment. This conservative approach preserved 94.2% of responses while strengthening the statistical evidence for metric preferences, with MDL significantly outperforming MI ($p = 0.67$, 95% Confidence Interval: $[0.56, 0.78]$). Even without the filtering MDL preference over MI was statistically significant.

# F BEST@K COMPRESSION IS A U-STATISTIC

We wish to estimate the expected compression ratio achieved by our *sample + rerank* method, which samples $k$ candidate refactorings, discards any that do not pass the tests, and selects the one with the lowest score (total log-prob).

**Background on U-Statistics.** Let $Z_1, \ldots, Z_n \overset{\text{i.i.d.}}{\sim} F$. For a symmetric function $h : \mathcal{Z}^k \to \mathbb{R}$, the *U-statistic of order $k$* is defined as

$$U_n = \binom{n}{k}^{-1} \sum_{1 \leq i_1 < \cdots < i_k \leq n} h(Z_{i_1}, \ldots, Z_{i_k}). \tag{6}$$

By construction,

$$\mathbb{E}[U_n] \;=\; \mathbb{E}[h(Z_1, \ldots, Z_k)], \tag{7}$$

so $U_n$ is an unbiased estimator of the population quantity $\theta = \mathbb{E}[h(Z_1, \ldots, Z_k)]$.

**Application to Best@k Compression.** Let each valid refactoring be a pair $Z = (S, C)$, where $S$ is the score and $C$ is the compression ratio. Define the symmetric function

$$h_k(z_1, \ldots, z_k) \;=\; C_{j^*}, \qquad j^* = \arg \min_{1 \le j \le k} S_j, \tag{8}$$

the compression ratio of the lowest-score refactoring among $k$ draws. The population target is then

$$\theta_k = \mathbb{E}[h_k(Z_1, \ldots, Z_k)]. \tag{9}$$

Given $n$ valid samples, our estimator is

$$\widehat{\theta}_k \;=\; \binom{n}{k}^{-1} \sum_{1 \le i_1 < \cdots < i_k \le n} h_k(Z_{i_1}, \ldots, Z_{i_k}). \tag{10}$$

**Proposition.** $\widehat{\theta}_k$ is a U-statistic of order $k$ with function $h_k$, and hence an unbiased estimator of $\theta_k$.

**Proof.** (1) *Symmetry of the function $h_k$.* $h_k$ selects the compression associated with the lowest score among its $k$ arguments. Permuting the inputs does not affect this outcome (ties can be resolved with a fixed, permutation-invariant rule). Thus $h_k$ is symmetric.

(2) *U-statistic form.* By definition, a U-statistic of order $k$ with kernel $h_k$ is

$$U_n = \binom{n}{k}^{-1} \sum_{1 \le i_1 < \cdots < i_k \le n} h_k(Z_{i_1}, \ldots, Z_{i_k}),$$

which matches $\widehat{\theta}_k$ exactly.

Therefore, $\widehat{\theta}_k$ is a U-statistic of order $k$. By the unbiasedness property of U-statistics,

$$\mathbb{E}[\widehat{\theta}_k] = \theta_k.$$

$\square$

Thus, our reported *best@k compression curves* provide unbiased estimates of the expected performance of the *sample + rerank* method.

## G  CLUSTERING ANALYSIS: CODECONTESTS

We analyze the coherence of the clusters underlying collections in MINICODE-**CodeContests**. In particular, we compare clustering based on o4-mini generated file descriptions against task descriptions. Since task descriptions in competition coding problems are designed to hide the algorithmic approach needed to solve problem, we expect that clusters based on file descriptions are more coherent. We use Normalized Tag Instance Entropy and Herfindahl-Hirschman Index to evaluate clusterings. Figure 8 shows our clustering approach yields more thematically coherent clusters, evidenced by achieving lower entropy and higher HHI values across the entire tested range of $N$. We provide definitions of our measures below.

### G.1  COLLECTION COHERENCE MEASURES

We use two measures to evaluate the thematic coherence of collections: Good collections should group files with a (1) concentrated and (2) identifiable set of shared *conceptual tags*, which for CodeContests are provided as ground truth (`trees`, `graphs`, etc.).

We provide the full definitions of the collection coherence measures below.

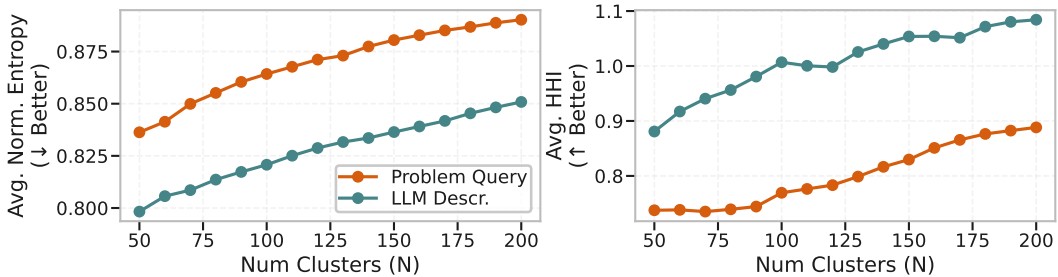

Figure 8: Clustering analysis of 4,596 Code Contest problems, comparing the thematic coherence of clusters formed using our proposed method versus REGAL-style clustering.

**Normalized Tag Instance Entropy:** This measures the concentration of tag *instances* within a collection $C$. Let $p_i$ be the proportion of the $i$-th unique tag type among all tag instances in $C$, and $D_C$ be the number of distinct tag types in $C$. If $D_C > 1$, the normalized entropy $H_N$ is defined as:

$$H_N = -\frac{\sum_{i=1}^{D_C} p_i \log_2 p_i}{\log_2 D_C} \tag{11}$$

If $D_C \leq 1$, then $H_N = 0$. Lower $H_N$ (closer to 0) indicates higher thematic purity, meaning fewer tag types dominate the bulk of tag mentions.

**Herfindahl-Hirschman Index (HHI) for Problem Presence:** This measures tag concentration across distinct *problems* in a cluster $C$. Let $s_t$ be the proportion of problems in $C$ that include tag $t$ (a problem contributes to $s_t$ if $t$ is one of its unique tags). A higher HHI signifies that the problems are collectively characterized by a smaller, more focused set of tags.

$$\text{HHI} = \sum_{t \in \text{Tags}(C)} s_t^2 \tag{12}$$

where $\text{Tags}(C)$ represents the set of unique tags present in cluster $C$.

## H    BENCHMARK COMPARISON

We compare our benchmark, MINICODE, to similar benchmarks in Table 4. We define creativity and design as the need to explore diverse solutions in order to find the best solution possible. For example, optimizing for program correctness alone does not require exploring a large solutions space, whereas optimizing a program for speed would. In the case of compressing large files, we must explore the large space of shared abstractions afforded by libraries in order to maximize compression.

Table 4: Comparison of Code Benchmarks

| Benchmark | Creativity/Design | Scale |
|---|---|---|
| SWE-bench (Jimenez et al., 2024) | Low | Repository |
| Commit-0 (Zhao et al., 2025) | Medium | Repository |
| RefactorBench (Gautam et al., 2025) | Low | File |
| ECCO (Waghjale et al., 2024) | High | Function |
| KernelBench (Ouyang et al., 2025) | High | Function |
| MINICODE(Ours) | High | Repository |

## I    FULL MINICODE CODECONTESTS RESULTS

We present the full agent scores for the CodeContests split in Table 5. The results are given both for each cluster of code sources, as well as averaged across clusters.

| Cluster | Agent | Tokens | CC | Pass % | MDL | MDL % |
|---|---|---|---|---|---|---|
| 0 | original | 9088 | 95 | 80.3 | 11745.85 | 100.0 |
| | sonnet 3.7 | 18114 | 176 | 87.0 | 15005.18 | 127.7 |
| | sonnet 4 | 11121 | 138 | 80.3 | 9901.53 | 84.3 |
| | codex-mini | 9321 | 95 | 80.3 | 9990.74 | 85.1 |
| 1 | original | 12531 | 255 | 89.7 | 13431.86 | 100.0 |
| | sonnet 3.7 | 10470 | 239 | 96.7 | 8933.65 | 66.5 |
| | sonnet 4 | 11325 | 298 | 96.7 | 8214.42 | 61.2 |
| | codex-mini | 12762 | 255 | 89.7 | 11798.73 | 87.8 |
| 2 | original | 14087 | 376 | 89.0 | 15012.77 | 100.0 |
| | sonnet 3.7 | 17345 | 429 | 91.3 | 13145.02 | 87.6 |
| | sonnet 4 | 14270 | 356 | 93.0 | 10522.66 | 70.1 |
| | codex-mini | 14318 | 376 | 89.0 | 13273.81 | 88.4 |
| 3 | original | 14261 | 246 | 90.3 | 13348.82 | 100.0 |
| | sonnet 3.7 | 20749 | 241 | 97.7 | 15859.02 | 118.8 |
| | sonnet 4 | 13433 | 197 | 80.7 | 11937.04 | 89.4 |
| | codex-mini | 14495 | 246 | 90.3 | 11616.41 | 87.0 |
| 4 | original | 17693 | 336 | 80.7 | 14665.16 | 100.0 |
| | sonnet 3.7 | 29860 | 358 | 100.0 | 20666.52 | 141.0 |
| | sonnet 4 | 18684 | 352 | 82.0 | 12801.21 | 87.3 |
| | codex-mini | 17923 | 336 | 80.7 | 12902.09 | 88.0 |
| 5 | original | 12588 | 286 | 92.0 | 12790.11 | 100.0 |
| | sonnet 3.7 | 10580 | 128 | 99.3 | 8435.12 | 65.9 |
| | sonnet 4 | 10416 | 155 | 99.3 | 9167.85 | 71.7 |
| | codex-mini | 12819 | 286 | 92.0 | 11086.19 | 86.7 |
| 6 | original | 11020 | 131 | 54.3 | 13540.41 | 100.0 |
| | sonnet 3.7 | 21747 | 502 | 88.0 | 19446.07 | 143.6 |
| | sonnet 4 | 11177 | 143 | 57.3 | 10492.00 | 77.5 |
| | codex-mini | 11251 | 131 | 54.3 | 11651.65 | 86.1 |
| 7 | original | 12301 | 180 | 80.0 | 12393.73 | 100.0 |
| | sonnet 3.7 | 16390 | 166 | 91.0 | 13371.59 | 107.9 |
| | sonnet 4 | 11625 | 150 | 85.7 | 9304.25 | 75.1 |
| | codex-mini | 12534 | 180 | 80.0 | 10549.04 | 85.1 |
| Avg | original | 12946 | 238 | 82.0 | 13366.09 | 100.0 |
| | sonnet 3.7 | 18157 | 280 | 93.9 | 14357.77 | 107.4 |
| | sonnet 4 | 12756 | 224 | 84.4 | 10292.62 | 77.1 |
| | codex-mini | 13178 | 238 | 82.0 | 11608.58 | 86.8 |

Table 5: Comparison of the pass rate and compression metrics of the original files, Claude Sonnet 4 and codex-mini refactorings across CodeContests clusters.

## J  REFACTORING EXAMPLES OF LIBRARIAN ON CODE CONTESTS

### J.1  EXAMPLE 1

In code snippets 3, 2, 5, 4 one example of 2 refactoring versions. Specifically, the versions are both passing at least as many test cases as the original and they have the biggest difference in MDL among all the sample refactorings for that tuple. Sample + rerank filtering selected refactoring V2. You can observe that refactoring V1 introduces some problem specific functions like `build_max_beauty_perm()`, while refactoring V2 sticks to more generally useful functions.

```
1  # ==== NEW HELPER FUNCTIONS ====
2  def compute_full_mask(i):
3      """Return mask of all 1s of the bit-length of i."""
4      return (1 << i.bit_length()) - 1
5
6  def build_max_beauty_perm(n):
7      """Build permutation of 0..n maximizing sum of i^p[i]."""
8      ans = [0] * (n + 1)
9      used = set()
10     for i in range(n, -1, -1):
11         if i in used:
12             continue
13         mask = compute_full_mask(i)
14         j = i ^ mask
15         ans[i], ans[j] = j, i
16         used.add(i)
17         used.add(j)
18     beauty = sum(i ^ ans[i] for i in range(n + 1))
19     return ans, beauty
20
21 def solve_xor_sum(u, v):
22     """
23     Find shortest array whose xor is u and sum is v.
24     Return list or None if impossible.
25     """
26     if u > v or (v - u) % 2:
27         return None
28     if u == v:
29         return [] if u == 0 else [u]
30     x = (v - u) // 2
31     # try two elements
32     if ((u + x) ^ x) == u:
33         return [u + x, x]
34     # fallback to three elements
35     return [u, x, x]
36
37 def build_trie(keys):
38     """
39     Build a binary trie with counts for 30-bit numbers.
40     Each node: [left_index, right_index, count].
41     """
42     tree = [[0, 0, 0]]
43     for x in keys:
44         now = 0
45         tree[now][2] += 1
46         for i in range(29, -1, -1):
47             b = (x >> i) & 1
48             if tree[now][b] == 0:
49                 tree[now][b] = len(tree)
50                 tree.append([0, 0, 0])
51             now = tree[now][b]
52             tree[now][2] += 1
53     return tree
54
55 def trie_pop_min_xor(tree, x):
56     """
57     Pop one key from trie to minimize x^key and return that minimal xor.
58     Decrements counts along the path.
59     """
60     now = 0
61     res = 0
62     for i in range(29, -1, -1):
63         b = (x >> i) & 1
64         nxt = tree[now][b]
65         if nxt and tree[nxt][2] > 0:
66             now = nxt
67         else:
68             now = tree[now][b ^ 1]
69             res |= (1 << i)
70         tree[now][2] -= 1
71     return res
```

Listing 2: Version 1, New Helpers

```
# ########## PROGRAM: node_16:cc_python_16 ##########

from codebank import *

def main():
    import sys
    data = sys.stdin.readline()
    if not data:
        return
    n = int(data)
    perm, beauty = build_max_beauty_perm(n)
    print(beauty)
    print(*perm)

if __name__ == "__main__":
    main()

# ########## PROGRAM: node_19:cc_python_19 ##########

from codebank import *

def main():
    import sys
    data = sys.stdin.readline
    n = int(data())
    A = list(map(int, data().split()))
    P = list(map(int, data().split()))
    trie = build_trie(P)
    O = [trie_pop_min_xor(trie, a) for a in A]
    print(*O)

if __name__ == "__main__":
    main()

# ########## PROGRAM: node_25:cc_python_25 ##########

from codebank import *

def main():
    import sys
    u, v = map(int, sys.stdin.readline().split())
    res = solve_xor_sum(u, v)
    if res is None:
        print(-1)
    else:
        print(len(res))
        if res:
            print(*res)

if __name__ == "__main__":
    main()
```

Listing 3: Version 1, Refactored Programs

```
# ==== NEW HELPER FUNCTIONS ====
def compute_complement(i):
    return i ^ ((1 << i.bit_length()) - 1)

def trie_add(trie, x, max_bit):
    trie[0][2] += 1
    now = 0
    for i in range(max_bit, -1, -1):
        bit = (x >> i) & 1
        if trie[now][bit] == 0:
            trie[now][bit] = len(trie)
            trie.append([0, 0, 0])
        now = trie[now][bit]
        trie[now][2] += 1

def trie_find_min_xor(trie, x, max_bit):
    now = 0
    ans = 0
    for i in range(max_bit, -1, -1):
        bit = (x >> i) & 1
        if trie[now][bit] and trie[trie[now][bit]][2] > 0:
            now = trie[now][bit]
        else:
            now = trie[now][bit ^ 1]
            ans |= (1 << i)
        trie[now][2] -= 1
    return ans
```

Listing 4: Version 2, New Helpers

```
1   # ########## PROGRAM: node_16:cc_python_16 ##########
2
3   from codebank import *
4
5   def main():
6       import sys
7       input = sys.stdin.readline
8       n = int(input())
9       ans = [-1] * (n + 1)
10      for i in range(n, -1, -1):
11          if ans[i] == -1:
12              z = compute_complement(i)
13              ans[i] = z
14              ans[z] = i
15      m = sum(i ^ ans[i] for i in range(n + 1))
16      print(m)
17      print(*ans)
18
19  if __name__ == "__main__":
20      main()
21
22  # ########## PROGRAM: node_19:cc_python_19 ##########
23
24  from codebank import *
25
26  def main():
27      import sys
28      input = sys.stdin.readline
29      n = int(input())
30      A = list(map(int, input().split()))
31      P = list(map(int, input().split()))
32      max_bit = max(max(A, default=0), max(P, default=0)).bit_length() - 1
33      trie = [[0, 0, 0]]
34      for x in P:
35          trie_add(trie, x, max_bit)
36      res = [trie_find_min_xor(trie, x, max_bit) for x in A]
37      print(*res)
38
39  if __name__ == "__main__":
40      main()
41
42  # ########## PROGRAM: node_25:cc_python_25 ##########
43
44  from codebank import *
45
46  def main():
47      u, v = map(int, input().split())
48      if u > v or ((v - u) & 1):
49          print(-1)
50      elif u == 0 and v == 0:
51          print(0)
52      elif u == v:
53          print(1)
54          print(u)
55      else:
56          w = (v - u) // 2
57          if (w & u) == 0:
58              d = u + w
59              print(2)
60              print(d, w)
61          else:
62              print(3)
63              print(u, w, w)
64
65  if __name__ == "__main__":
66      main()
```

Listing 5: Version 2, Refactored Programs

## J.2 EXAMPLE 2

In code snippets 7, 6, 9, 8 is another example of 2 refactorings where V1 was better according to LIBRARIAN. We can observe that V2 creates helper functions that are overly specific to the problem. You can see that refactoring V2 introduces overly specialized functions like `dijkstra_special()` or `compute_min_moves_opposite_parity()`. In comparison, refactoring V1 generates only general versions of these functions (e.g. `dijkstra()`).

```python
# ==== NEW HELPER FUNCTIONS ====
def read_ints():
    return list(map(int, input().split()))

def build_adj_undirected(n, edges):
    adj = [[] for _ in range(n)]
    for u, v, w in edges:
        adj[u].append((v, w))
        adj[v].append((u, w))
    return adj

def dijkstra(adj, src):
    from heapq import heappush, heappop
    INF = 10**18
    n = len(adj)
    dist = [INF]*n
    parent = [-1]*n
    dist[src] = 0
    heap = [(0, src)]
    while heap:
        d, u = heappop(heap)
        if d > dist[u]:
            continue
        for v, w in adj[u]:
            nd = d + w
            if nd < dist[v]:
                dist[v] = nd
                parent[v] = u
                heappush(heap, (nd, v))
    return dist, parent

def reconstruct_path(parent, dest):
    path = []
    u = dest
    while u != -1:
        path.append(u+1)
        u = parent[u]
    return path[::-1]

def multi_source_bfs(neighbors, sources):
    from collections import deque
    n = len(neighbors)
    dist = [-1]*n
    dq = deque()
    for u in sources:
        if dist[u] == -1:
            dist[u] = 0
            dq.append(u)
    while dq:
        u = dq.popleft()
        for v in neighbors[u]:
            if dist[v] == -1:
                dist[v] = dist[u] + 1
                dq.append(v)
    return dist
```

Listing 6: Version 1, New Helpers

```
1   # ########## PROGRAM: node_16:cc_python_16 ##########
2
3   from codebank import *
4
5   def main():
6       import heapq
7       n, m = read_ints()
8       edges = [(u-1, v-1, w) for u, v, w in (read_ints() for _ in range(m))]
9       adj = build_adj_undirected(n, edges)
10      INF = 10**20
11      dist = [INF]*n
12      dist[0] = 0
13      last_w = [0]*n
14      heap = [(0, 0)]
15      while heap:
16          d, u = heapq.heappop(heap)
17          if d > dist[u]:
18              continue
19          # record last edges
20          for v, w in adj[u]:
21              last_w[v] = w
22          # expand two-edge moves
23          for v, w1 in adj[u]:
24              tw = last_w[v]
25              for x, w2 in adj[v]:
26                  nd = d + (tw + w2)**2
27                  if nd < dist[x]:
28                      dist[x] = nd
29                      heapq.heappush(heap, (nd, x))
30      out = []
31      for x in dist:
32          out.append(str(x if x < INF else -1))
33      print(" ".join(out))
34
35  if __name__ == "__main__":
36      main()
37
38  # ########## PROGRAM: node_17:cc_python_17 ##########
39
40  from codebank import *
41
42  def main():
43      n, m = read_ints()
44      edges = [(u-1, v-1, w) for u, v, w in (read_ints() for _ in range(m))]
45      adj = build_adj_undirected(n, edges)
46      dist, parent = dijkstra(adj, 0)
47      if dist[n-1] >= 10**18:
48          print(-1)
49      else:
50          path = reconstruct_path(parent, n-1)
51          print(*path)
52
53  if __name__ == "__main__":
54      main()
55
56  # ########## PROGRAM: node_19:cc_python_19 ##########
57
58  from codebank import *
59
60  def main():
61      n = int(input())
62      a = read_ints()
63      # build reversed graph: for each move i->j, add edge j->i
64      neighbors = [[] for _ in range(n)]
65      for i, val in enumerate(a):
66          for j in (i - val, i + val):
67              if 0 <= j < n:
68                  neighbors[j].append(i)
69      # BFS from all even and all odd positions separately
70      even_sources = [i for i, val in enumerate(a) if val % 2 == 0]
71      odd_sources  = [i for i, val in enumerate(a) if val % 2 == 1]
72      dist_even = multi_source_bfs(neighbors, even_sources)
73      dist_odd  = multi_source_bfs(neighbors, odd_sources)
74      # for odd a[i], answer is dist to nearest even => dist_even; else dist_odd
75      ans = [dist_even[i] if a[i] % 2 == 1 else dist_odd[i] for i in range(n)]
76      print(*ans)
77
78  if __name__ == "__main__":
79      main()
```

Listing 7: Version 1, Refactored Programs

```python
#
# ==== NEW HELPER FUNCTIONS ====
def read_ints():
    return list(map(int, input().split()))

def build_undirected_weighted_graph(n, m):
    from collections import defaultdict
    adj = defaultdict(list)
    for _ in range(m):
        u, v, w = read_ints()
        u -= 1; v -= 1
        adj[u].append((v, w))
        adj[v].append((u, w))
    return adj

def dijkstra(adj, src, n):
    import heapq
    INF = 10**18
    dist = [INF]*n
    parent = [-1]*n
    visited = [False]*n
    dist[src] = 0
    heap = [(0, src)]
    while heap:
        d, u = heapq.heappop(heap)
        if visited[u]:
            continue
        visited[u] = True
        for v, w in adj.get(u, ()):
            nd = d + w
            if nd < dist[v]:
                dist[v] = nd
                parent[v] = u
                heapq.heappush(heap, (nd, v))
    return dist, parent

def reconstruct_path(parent, dest):
    path = []
    while dest != -1:
        path.append(dest+1)
        dest = parent[dest]
    return path[::-1]

def dijkstra_special(e, n, src):
    import heapq
    INF = 10**18
    d = [INF]*n
    d[src] = 0
    heap = [(0, src)]
    while heap:
        cd, v = heapq.heappop(heap)
        if cd > d[v]:
            continue
        td = {}
        for u, w in e.get(v, ()):
            td[u] = w
        for u, w1 in td.items():
            for x, w2 in e.get(u, ()):
                cost = cd + (w1 + w2)**2
                if cost < d[x]:
                    d[x] = cost
                    heapq.heappush(heap, (cost, x))
    return d

def compute_min_moves_opposite_parity(a):
    from collections import deque
    n = len(a)
    go = [[] for _ in range(n)]
    ans = [-1]*n
    q = deque()
    for i, val in enumerate(a):
        for j in (i - val, i + val):
            if 0 <= j < n:
                if (a[j] % 2) != (val % 2):
                    ans[i] = 1
                    q.append(i)
                    break
                else:
                    go[j].append(i)
    while q:
        u = q.popleft()
        for v in go[u]:
            if ans[v] == -1:
                ans[v] = ans[u] + 1
                q.append(v)
    return ans
```

Listing 8: Version 2, New Helpers

```
1   #
2   # ########## PROGRAM: node_16:cc_python_16 ##########
3
4   from codebank import *
5
6   def main():
7       n, m = read_ints()
8       e = {}
9       for _ in range(m):
10          u, v, w = read_ints()
11          u -= 1; v -= 1
12          e.setdefault(u, []).append((v, w))
13          e.setdefault(v, []).append((u, w))
14      d = dijkstra_special(e, n, 0)
15      print(" ".join(str(-1 if x >= 10**18 else int(x)) for x in d))
16
17  if __name__ == "__main__":
18      main()
19
20  # ########## PROGRAM: node_17:cc_python_17 ##########
21
22  from codebank import *
23
24  def main():
25      n, m = read_ints()
26      adj = build_undirected_weighted_graph(n, m)
27      dist, parent = dijkstra(adj, 0, n)
28      if dist[n-1] >= 10**18:
29          print(-1)
30      else:
31          path = reconstruct_path(parent, n-1)
32          print(" ".join(map(str, path)))
33
34  if __name__ == "__main__":
35      main()
36
37  # ########## PROGRAM: node_19:cc_python_19 ##########
38
39  from codebank import *
40
41  def main():
42      n = int(input())
43      a = read_ints()
44      ans = compute_min_moves_opposite_parity(a)
45      print(" ".join(map(str, ans)))
46
47  if __name__ == "__main__":
48      main()
```

Listing 9: Version 2, Refactored Programs

## K  OBFUSCATION EXAMPLE

We provide the full source code, both refactored and obfuscated, for the MDL and token comparison here.

Listing 10: Refactored code from modeling_llama.py from the refactored Transformers repository.

```python
from typing import Optional, Union

import torch
from torch import nn

from ...cache_utils import Cache, DynamicCache
from ...generation import GenerationMixin
from ...masking_utils import create_causal_mask
from ...modeling_layers import (
    GenericForQuestionAnswering,
    GenericForSequenceClassification,
    GenericForTokenClassification,
)
from ...modeling_outputs import (
    BaseModelOutputWithPast,
    CausalLMOutputWithPast,
)
from ...modeling_utils import PreTrainedModel
from ...processing_utils import Unpack
from ...utils import TransformersKwargs, auto_docstring, can_return_tuple, logging
from ...utils.generic import check_model_inputs
from .configuration_llama import LlamaConfig

from ..shared_library import (
```

```
1350        rotate_half,
1351        apply_rotary_pos_emb,
1352        repeat_kv,
1353        eager_attention_forward,
1354        RMSNorm,
1355        BaseMLP,
1356        BaseRotaryEmbedding,
1357        BaseAttention,
1358        BaseDecoderLayer,
        )
1359

1360
        logger = logging.get_logger(__name__)
1361

1362
1363    class LlamaRMSNorm(RMSNorm):
1364        pass
1365

1366    class LlamaRotaryEmbedding(BaseRotaryEmbedding):
1367        pass
1368

1369    class LlamaMLP(BaseMLP):
1370        def __init__(self, config):
1371            super().__init__(config, mlp_bias=config.mlp_bias)
1372

1373    class LlamaAttention(BaseAttention):
1374        def __init__(self, config: LlamaConfig, layer_idx: int):
1375            super().__init__(
1376                config=config,
1377                layer_idx=layer_idx,
1378                attention_bias=config.attention_bias,
1379                sliding_window=None
1380            )
1381

1382    class LlamaDecoderLayer(BaseDecoderLayer):
1383        def __init__(self, config: LlamaConfig, layer_idx: int):
1384            super().__init__(
1385                config=config,
1386                layer_idx=layer_idx,
1387                norm_class=LlamaRMSNorm,
1388                mlp_class=LlamaMLP,
1389                attention_class=LlamaAttention
1390            )
1391

1392    @auto_docstring
1393    class LlamaPreTrainedModel(PreTrainedModel):
1394        config: LlamaConfig
        base_model_prefix = "model"
1395        supports_gradient_checkpointing = True
1396        _no_split_modules = ["LlamaDecoderLayer"]
1397        _skip_keys_device_placement = ["past_key_values"]
1398        _supports_flash_attn = True
        _supports_sdpa = True
1399        _supports_flex_attn = True

1400        _can_compile_fullgraph = True
1401        _supports_attention_backend = True
1402        _can_record_outputs = {
            "hidden_states": LlamaDecoderLayer,
1403            "attentions": LlamaAttention,
        }

    @auto_docstring
    class LlamaModel(LlamaPreTrainedModel):
        def __init__(self, config: LlamaConfig):
            super().__init__(config)
            self.padding_idx = config.pad_token_id
            self.vocab_size = config.vocab_size

            self.embed_tokens = nn.Embedding(config.vocab_size, config.hidden_size, self.padding_idx)
            self.layers = nn.ModuleList(
                [LlamaDecoderLayer(config, layer_idx) for layer_idx in range(config.num_hidden_layers)]
            )
            self.norm = LlamaRMSNorm(config.hidden_size, eps=config.rms_norm_eps)
            self.rotary_emb = LlamaRotaryEmbedding(config=config)
```

```python
            self.gradient_checkpointing = False

            self.post_init()

    @check_model_inputs
    @auto_docstring
    def forward(
        self,
        input_ids: Optional[torch.LongTensor] = None,
        attention_mask: Optional[torch.Tensor] = None,
        position_ids: Optional[torch.LongTensor] = None,
        past_key_values: Optional[Cache] = None,
        inputs_embeds: Optional[torch.FloatTensor] = None,
        cache_position: Optional[torch.LongTensor] = None,
        use_cache: Optional[bool] = None,
        **kwargs: Unpack[TransformersKwargs],
    ) -> BaseModelOutputWithPast:
        if (input_ids is None) ^ (inputs_embeds is not None):
            raise ValueError("You must specify exactly one of input_ids or inputs_embeds")

        if inputs_embeds is None:
            inputs_embeds: torch.Tensor = self.embed_tokens(input_ids)

        if use_cache and past_key_values is None:
            past_key_values = DynamicCache(config=self.config)

        if cache_position is None:
            past_seen_tokens = past_key_values.get_seq_length() if past_key_values is not None else 0
            cache_position: torch.Tensor = torch.arange(
                past_seen_tokens, past_seen_tokens + inputs_embeds.shape[1], device=inputs_embeds.device
            )

        if position_ids is None:
            position_ids = cache_position.unsqueeze(0)

        causal_mask = create_causal_mask(
            config=self.config,
            input_embeds=inputs_embeds,
            attention_mask=attention_mask,
            cache_position=cache_position,
            past_key_values=past_key_values,
            position_ids=position_ids,
        )

        hidden_states = inputs_embeds
        position_embeddings = self.rotary_emb(hidden_states, position_ids)

        for decoder_layer in self.layers[: self.config.num_hidden_layers]:
            hidden_states = decoder_layer(
                hidden_states,
                attention_mask=causal_mask,
                position_ids=position_ids,
                past_key_values=past_key_values,
                cache_position=cache_position,
                position_embeddings=position_embeddings,
                **kwargs,
            )

        hidden_states = self.norm(hidden_states)
        return BaseModelOutputWithPast(
            last_hidden_state=hidden_states,
            past_key_values=past_key_values,
        )

@auto_docstring
class LlamaForCausalLM(LlamaPreTrainedModel, GenerationMixin):
    _tied_weights_keys = ["lm_head.weight"]
    _tp_plan = {"lm_head": "colwise_rep"}
    _pp_plan = {"lm_head": (["hidden_states"], ["logits"])}

    def __init__(self, config):
        super().__init__(config)
        self.model = LlamaModel(config)
        self.vocab_size = config.vocab_size
        self.lm_head = nn.Linear(config.hidden_size, config.vocab_size, bias=False)

        self.post_init()

    def set_decoder(self, decoder):
        self.model = decoder
```

```
1458
1459        def get_decoder(self):
1460            return self.model
1461
1462        @can_return_tuple
1463        @auto_docstring
1464        def forward(
1465            self,
1466            input_ids: Optional[torch.LongTensor] = None,
1467            attention_mask: Optional[torch.Tensor] = None,
1468            position_ids: Optional[torch.LongTensor] = None,
1469            past_key_values: Optional[Cache] = None,
1470            inputs_embeds: Optional[torch.FloatTensor] = None,
1471            labels: Optional[torch.LongTensor] = None,
1472            use_cache: Optional[bool] = None,
1473            cache_position: Optional[torch.LongTensor] = None,
1474            logits_to_keep: Union[int, torch.Tensor] = 0,
1475            **kwargs: Unpack[TransformersKwargs],
1476        ) -> CausalLMOutputWithPast:
1477            r"""
1478            Example:
1479
1480            ```python
1481            >>> from transformers import AutoTokenizer, LlamaForCausalLM
1482
1483            >>> model = LlamaForCausalLM.from_pretrained("meta-llama/Llama-2-7b-hf")
1484            >>> tokenizer = AutoTokenizer.from_pretrained("meta-llama/Llama-2-7b-hf")
1485
1486            >>> prompt = "Hey, are you conscious? Can you talk to me?"
1487            >>> inputs = tokenizer(prompt, return_tensors="pt")
1488
1489            >>> # Generate
1490            >>> generate_ids = model.generate(inputs.input_ids, max_length=30)
1491            >>> tokenizer.batch_decode(generate_ids, skip_special_tokens=True, clean_up_tokenization_spaces=False)[0]
1492            "Hey, are you conscious? Can you talk to me?\nI'm not conscious, but I can talk to you."
1493            ```"""
1494            outputs: BaseModelOutputWithPast = self.model(
1495                input_ids=input_ids,
1496                attention_mask=attention_mask,
1497                position_ids=position_ids,
1498                past_key_values=past_key_values,
1499                inputs_embeds=inputs_embeds,
1500                use_cache=use_cache,
1501                cache_position=cache_position,
1502                **kwargs,
1503            )
1504
1505            hidden_states = outputs.last_hidden_state
1506            slice_indices = slice(-logits_to_keep, None) if isinstance(logits_to_keep, int) else logits_to_keep
1507            logits = self.lm_head(hidden_states[:, slice_indices, :])
1508
1509            loss = None
1510            if labels is not None:
1511                loss = self.loss_function(logits=logits, labels=labels, vocab_size=self.config.vocab_size, **kwargs)

            return CausalLMOutputWithPast(
                loss=loss,
                logits=logits,
                past_key_values=outputs.past_key_values,
                hidden_states=outputs.hidden_states,
                attentions=outputs.attentions,
            )

class LlamaForSequenceClassification(GenericForSequenceClassification, LlamaPreTrainedModel): ...

class LlamaForQuestionAnswering(GenericForQuestionAnswering, LlamaPreTrainedModel):
    base_model_prefix = "transformer"

class LlamaForTokenClassification(GenericForTokenClassification, LlamaPreTrainedModel): ...

__all__ = [
    "LlamaForCausalLM",
    "LlamaModel",
    "LlamaPreTrainedModel",
    "LlamaForSequenceClassification",
    "LlamaForQuestionAnswering",
    "LlamaForTokenClassification",
```

1512
1513
1514
1515

Listing 11: Obfuscated code from modeling_llama.py from the refactored Transformers repository.

```python
from typing import Optional as O,Union as U
import torch as T
from torch import nn as N
from ...cache_utils import Cache as C,DynamicCache as D
from ...generation import GenerationMixin as G
from ...masking_utils import create_causal_mask as m
from ...modeling_layers import GenericForQuestionAnswering as Q,GenericForSequenceClassification as S,
    GenericForTokenClassification as K
from ...modeling_outputs import BaseModelOutputWithPast as B,CausalLMOutputWithPast as L
from ...modeling_utils import PreTrainedModel as P
from ...processing_utils import Unpack as X
from ...utils import TransformersKwargs as W,auto_docstring as a,can_return_tuple as r,logging
from ...utils.generic import check_model_inputs as c
from .configuration_llama import LlamaConfig as F
from ..shared_library import rotate_half,apply_rotary_pos_emb,repeat_kv,eager_attention_forward,RMSNorm as R,
    BaseMLP as M,BaseRotaryEmbedding as E,BaseAttention as A,BaseDecoderLayer as Y
l=logging.get_logger(__name__)
class Z(R):pass
class I(E):pass
class J(M):
 def __init__(s,g):super().__init__(g,mlp_bias=g.mlp_bias)
class H(A):
 def __init__(s,g:F,i:int):super().__init__(config=g,layer_idx=i,attention_bias=g.attention_bias,sliding_window=
     None)
class V(Y):
 def __init__(s,g:F,i:int):super().__init__(config=g,layer_idx=i,norm_class=Z,mlp_class=J,attention_class=H)
@a
class o(P):
 config:F
 base_model_prefix="model"
 supports_gradient_checkpointing=True
 _no_split_modules=["LlamaDecoderLayer"]
 _skip_keys_device_placement=["past_key_values"]
 _supports_flash_attn=True
 _supports_sdpa=True
 _supports_flex_attn=True
 _can_compile_fullgraph=True
 _supports_attention_backend=True
 _can_record_outputs={"hidden_states":V,"attentions":H}
@a
class u(o):
 def __init__(s,g:F):
  super().__init__(g)
  s.padding_idx=g.pad_token_id
  s.vocab_size=g.vocab_size
  s.embed_tokens=N.Embedding(g.vocab_size,g.hidden_size,s.padding_idx)
  s.layers=N.ModuleList([V(g,i)for i in range(g.num_hidden_layers)])
  s.norm=Z(g.hidden_size,eps=g.rms_norm_eps)
  s.rotary_emb=I(config=g)
  s.gradient_checkpointing=False
  s.post_init()
 @c
 @a
 def forward(s,input_ids:O[T.LongTensor]=None,attention_mask:O[T.Tensor]=None,position_ids:O[T.LongTensor]=None,
     past_key_values:O[C]=None,inputs_embeds:O[T.FloatTensor]=None,cache_position:O[T.LongTensor]=None,use_cache:
     O[bool]=None,**k:X[W])->B:
  if(input_ids is None)^(inputs_embeds is not None):raise ValueError("You must specify exactly one of input_ids
      or inputs_embeds")
  if inputs_embeds is None:inputs_embeds:T.Tensor=s.embed_tokens(input_ids)
  if use_cache and past_key_values is None:past_key_values=D(config=s.config)
  if cache_position is None:
   p=past_key_values.get_seq_length()if past_key_values is not None else 0
   cache_position:T.Tensor=T.arange(p,p+inputs_embeds.shape[1],device=inputs_embeds.device)
  if position_ids is None:position_ids=cache_position.unsqueeze(0)
  f=m(config=s.config,input_embeds=inputs_embeds,attention_mask=attention_mask,cache_position=cache_position,
      past_key_values=past_key_values,position_ids=position_ids)
  h=inputs_embeds
  e=s.rotary_emb(h,position_ids)
  for d in s.layers[:s.config.num_hidden_layers]:h=d(h,attention_mask=f,position_ids=position_ids,past_key_values
      =past_key_values,cache_position=cache_position,position_embeddings=e,**k)
  h=s.norm(h)
  return B(last_hidden_state=h,past_key_values=past_key_values)
@a
class t(o,G):
 _tied_weights_keys=["lm_head.weight"]
 _tp_plan={"lm_head":"colwise_rep"}
```

```
_pp_plan={"lm_head":(["hidden_states"],["logits"])}
def __init__(s,g):
 super().__init__(g)
 s.model=u(g)
 s.vocab_size=g.vocab_size
 s.lm_head=N.Linear(g.hidden_size,g.vocab_size,bias=False)
 s.post_init()
def set_decoder(s,d):s.model=d
def get_decoder(s):return s.model
@r
@a
def forward(s,input_ids:O[T.LongTensor]=None,attention_mask:O[T.Tensor]=None,position_ids:O[T.LongTensor]=None,
     past_key_values:O[C]=None,inputs_embeds:O[T.FloatTensor]=None,labels:O[T.LongTensor]=None,use_cache:O[bool
     ]=None,cache_position:O[T.LongTensor]=None,logits_to_keep:U[int,T.Tensor]=0,**k:X[W])->L:
 o:B=s.model(input_ids=input_ids,attention_mask=attention_mask,position_ids=position_ids,past_key_values=
     past_key_values,inputs_embeds=inputs_embeds,use_cache=use_cache,cache_position=cache_position,**k)
 h=o.last_hidden_state
 i=slice(-logits_to_keep,None)if isinstance(logits_to_keep,int)else logits_to_keep
 g=s.lm_head(h[:,i,:])
 n=None
 if labels is not None:n=s.loss_function(logits=g,labels=labels,vocab_size=s.config.vocab_size,**k)
 return L(loss=n,logits=g,past_key_values=o.past_key_values,hidden_states=o.hidden_states,attentions=o.
     attentions)
class b(S,o):...
class x(Q,o):base_model_prefix="transformer"
class y(K,o):...
__all__=["t","u","o","b","x","y"]
```