# OpenReview forum: "Refactoring Codebases Through Library Design"
_ICLR.cc/2026/Conference — Submitted to ICLR 2026_

### Official Review · Reviewer_p9FN · 2025-10-28

**Soundness:** 4
**Presentation:** 3
**Contribution:** 3
**Rating:** 10
**Confidence:** 2

**Summary:**

The paper introduces MINICODE, an open-ended benchmark for refactoring multiple source files into a reusable library, and LIBRARIAN, a sample-and-rerank approach to synthesize such libraries. A comparative study finds that Minimum Description Length aligns best with developer preferences for refactoring quality; the method is further validated on real-world repositories, showing promising practical implications.

**Strengths:**

+ The investigated problem is clearly defined, novel and important to the community. MINICODE emphasizes open-ended library design, objective verifiability via unit tests, and large-context understanding across multiple files, addressing gaps in prior repo-level benchmarks.
+ LIBRARIAN combines sample-and-rerank with semantic clustering and a progressive, cross-cluster library accumulation strategy, which is practical for long-context constraints.
+ The paper compares Tokens/MDL vs. CC/MI, shows MDL better promotes reusable abstractions, and corroborates this with a human study.
+ Evaluations show strong empirical results. LIBRARIAN shows above 90% pass rate on CodeContests, which is a successful practical application of library learning to real software projects.

**Weaknesses:**

- In practice, a single file often mixes heterogeneous concerns (utility helpers, adapters, domain logic, classes), so the reviewer is not sure that file-level clusters could be too coarse.
- Figure 6 is a case study refactoring the code from the HuggingFace Transformers codebase. However, it is a little complex to understand. Authors are suggested to walk through the end-to-end pipeline that produced this figure.

**Questions:**

No question.

---

> ### Author Response · Authors · 2025-11-21
>
> We thank the reviewer for their valuable time and feedback, which has helped us refine and improve the clarity of our work. We have addressed each point raised below and hope our clarifications resolve any concerns.
>
>
> > …single file often mixes heterogeneous concerns…not sure that file-level clusters could be too coarse.
>
> This is a fair concern, especially when the LLM is unable to practically process full files. We’ve revised to mention this in limitations and future work:
>
> “Our clustering algorithm groups related files, but cannot split files into pieces to make more targeted clusters, which could prove important for long files that mix-and-match different concerns. In principle, more fine-grained clustering could address this, or alternatively, advances in the backend LLM could allow processing full files mixing many elements.”
>
> > Fig 6… a little complicated…suggested to walk through the end-to-end pipeline that produced this figure.
>
> We will add a walk-through explanation of the how Figure 6 was obtained—and what LIBRARIAN did to refactored the example—in the Appendix.

---

### Official Review · Reviewer_aB2G · 2025-10-31

**Soundness:** 2
**Presentation:** 2
**Contribution:** 3
**Rating:** 4
**Confidence:** 4

**Summary:**

This paper studies code refactoring in software engineering, focusing on maintainability and reusability. The authors analyze different metrics for evaluating refactoring quality and find that Minimum Description Length (MDL) is the best metric. To evaluate and advance research on code refactoring, the paper introduces the MINICODE benchmark and the LIBRARIAN method.

**Strengths:**

- MINICODE offers a practical and effective benchmark for evaluating code refactoring
- The proposed metric MDL is reasonable and interesting.

**Weaknesses:**

- The proposed sample-and-rerank approach is relatively simple, and the methodological insights it provides are limited.
- The main risk of using MDL is that it can be heavily influenced by a single model. The paper only briefly discusses cross-model agreement for MDL in Section 6; a more detailed analysis would make the claim more convincing.
- Even if the refactored code passes all unit tests, there is still a risk of semantic inequivalence with the original code. The paper lacks an analysis of this risk.

**Questions:**

- Are there any cases where refactorings produced by LIBRARIAN in real projects have been successfully merged into the community codebase?
- Typos: line 218: “set of set of”; line 273: “more more.”
- Missing citation for the MI metric.
- What does “change from non-refactored” mean in Figures 2 and 3, and how is it calculated?

---

> ### Author Response · Authors · 2025-11-21
>
> We thank the reviewer for their valuable time and feedback. We have addressed each point raised below and hope our clarifications resolve any concerns.
>
>
> > …sample-and-rerank approach is relatively simple…insights it provides are limited
>
> The core insights are what to rank by, and combining sampling/search with clustering. Neither insight was previously studied in the (long) history of library learning, and we think it’s good that a simple method achieves state of the art!
>
> > …MDL…can be heavily influenced by a single model… a more detailed analysis would make the claim more convincing.
>
> Thanks for suggesting this analysis. Across five models (Qwen, Llama, DeepSeek-Coder, Mistral, OLMo2), we get $\bf{r=0.965}$ correlation for cross-model likelihood predictions, so MDL is not heavily dependent on the backend LLM. The initial submission only considered two models (line 251), and we’ll revise to add these results on 5 models.
>
>
> > Even if the refactored code passes all unit tests… there is still a risk of semantic inequivalence with the original code…lacks analysis of this risk
>
> This is a fair concern; we will note it explicitly in the revised paper. That said, our Transformers and Diffusers refactorings are validated against the same integration test suites the HuggingFace authors use to decide whether a code change is safe to merge. So while not a formal proof, these tests provide a strong practical proxy for semantic equivalence.
>
>
> > What does “change from non-refactored” mean in Fig 2 and 3, and how is it calculated?
>
> “Change from non-refactored’’ measures how much a metric improves relative to the original (unrefactored) files, using the Best@k estimator. For each cluster, we sample (n) refactorings and consider every subset of size ($k$). For each subset we take the best refactoring under metric ($M$), compute its percentage change from the original
>  $$
>  \Delta = \frac{M_{\text{best@}k}-M_{\text{original}}}{M_{\text{original}}}\times 100,
>  $$
>  and then average these deltas over all subsets of size $k$, and finally over all clusters. This gives the expected improvement one should get when sampling ($k$) candidates, making Figures 2–3 comparable across metrics.
>
>
>
> > Missing citation for the MI metric
>
> Thanks for pointing this out. We will be adding the following citation to the paper:
>
> ```tex
> @INPROCEEDINGS{242525,
>   author={Oman, P. and Hagemeister, J.},
>   booktitle={Proceedings Conference on Software Maintenance 1992},
>   title={Metrics for assessing a software system's maintainability},
>   year={1992},
>   volume={},
>   number={},
>   pages={337-344},
>   keywords={Software maintenance;Software systems;Software measurement;Tree data structures;Documentation;Taxonomy;Environmental management;Lifting equipment;Software engineering;Software testing},
>   doi={10.1109/ICSM.1992.242525}}
> ```

---

### Official Review · Reviewer_vBJz · 2025-10-31

**Soundness:** 3
**Presentation:** 3
**Contribution:** 2
**Rating:** 4
**Confidence:** 4

**Summary:**

This paper addresses the problem of refactoring existing code to generate library code that is then used for rewriting the original code snippets. The paper assembles a benchmark called MiniCode and proposes a technique called Librarian. The paper compares different ranking methods and finds that minimum description length (MDL) to be a suitable metric to rank different refactoring suggestions. The evaluation on MiniCode shows effectiveness of Librarian.

**Strengths:**

- Code refactoring is an important software engineering activity. This paper demonstrates progress on this problem using a pipeline of clustering of code by natural language summary, cluster-specific library extraction and then rewriting the complete code corpus.
- It assembles a benchmark taking code contest solutions, previous refactoring benchmarks and small sets of related files from transformers and diffusers libraries. The resulting refactorings are ranked using MDL and evaluated for correctness using tests. It reports better refactoring accuracy than closely related work Regal on a subset of the benchmark.
- The comparative analysis between MDL, tokens and software engineering metrics is interesting and justifies the use of MDL in ranking.

**Weaknesses:**

- While the problem of refactoring is important, the proposed method is evaluated in limited setting. It does not present results at large scale where refactorings are most important and useful. Though the paper states that the proposed method is evaluated on "real-world code bases", the scope is restricted to a total of 3 tasks with 10 files each from 2 repositories.
- The paper's novelty over past work, Regal, is limited as both of them apply clustering based refactoring.
- The study of different metrics showing MDL > tokens > MI and that test-time scaling can help are the interesting parts in the paper. As an aside, I could find a citation for the MI metric; please add.

**Questions:**

- In coding contest setting, the same problem can be solved using different algorithmic techniques. How well does the clustering work in such a setting? What is the quality of clustering if you were give entire code repositories (e.g., transformers) rather than hand-selected 10 files?
- As noted in the experiments, Librarian can create functions that are used only once. Curious if they are used within the refactored code itself (e.g., private methods) or outside?
- How does the performance vary if you use GPT models instead of o4-mini on CodeContents?

---

> ### Author Response · Authors · 2025-11-21
>
> We thank the reviewer for their valuable time and feedback. We have addressed each point raised below and hope our clarifications resolve any concerns.
>
> > …evaluated in a limited setting…states evaluated on “real-world-codebases”
>
> Librarian refactors 10 Transformers and 11 Diffusers files (total 12,915 lines of code). But our initial submission overstated the extent to which LIBRARIAN is ready for deployment on full repos. We simply hoped to say that our work is head-and-shoulders closer to real-world use relative to past library learning algorithms, which worked with 10-20 lines of code, and almost never human written programs. Additionally, cost is a limitation in applying these methods.
>
> We’ve revised to say the following contributions: LIBRARIAN “outperforms prior library learners on existing benchmarks and scales to real-world human-written Python code” (previously “scales to complex codebases”) and gives “refactorings of real-world Python sources… successfully refactors files from the Huggingface Transformers and Diffusers libraries” (previously “refactorings of real-world codebases… successfully refactors the Transformers and Diffusers libraries”). We still say that “to the best of our knowledge this is the first time library learning has been successfully applied to real-world software projects” – because indeed this is true!
>
> > The scope is restricted to a total of 3 tasks with 10 files each from 2 repositories.
>
> The main limiting factor in scale is cost. We estimate the cost of a single experiment for refactoring 1 Transformer task (5 files) with $k=15$ samples to cost around 100 dollars.
>
> >  novelty over past work...is limited...
>
> The core insights are what to rank by, and how to combine sampling/search with clustering. Neither insight was previously studied in the long history of library learning, and we think it’s a positive result that applying them allows such a simple method to achieve state of the art.
>
> > In coding contest settings, the same problem can be solved with different algorithmic techniques. How well does clustering work there?
>
> Yes—clustering handles this because we cluster by descriptions of how a program solves a problem, not by the problem statement itself. This groups related techniques (e.g., BFS vs. Dijkstra), so solutions with similar algorithms cluster together even if their goals differ. Past work (i.e. Regal) instead cluster by problem description / goal, which unlike LIBRARIAN, does not as consistently yield clusters amenable to refactoring.
>
> > Regarding functions noted as "used only once": "Curious if they are used within the refactored code itself (e.g., private methods) or outside?"
>
> These are library functions that ended up being used by only one refactored program. In CodeContests this is expected: some solutions use very specific subroutines that get extracted but don’t appear elsewhere. The key trend is that as the library grows, the fraction of single-use functions falls while total abstractions rise—so today’s single-use function becomes multi-use as more files are added.
>
> > …GPT models instead of o4-mini on CodeContests?
>
> We tried GPT 3.5-turbo and GPT4, and found qualitatively similar results. We’ll add it to the appendix.
>
> > ...MI metric; please add.
>
> Thanks for pointing this out. We will be adding the following citation to the paper:
>
> ```tex
> @INPROCEEDINGS{242525,
>   author={Oman, P. and Hagemeister, J.},
>   booktitle={Proceedings Conference on Software Maintenance 1992},
>   title={Metrics for assessing a software system's maintainability},
>   year={1992},
>   volume={},
>   number={},
>   pages={337-344},
>   keywords={Software maintenance;Software systems;Software measurement;Tree data structures;Documentation;Taxonomy;Environmental management;Lifting equipment;Software engineering;Software testing},
>   doi={10.1109/ICSM.1992.242525}}
> ```

---

> > ### Comment · Reviewer_vBJz · 2025-11-27
> > **Thank you**
> >
> > Thank you for rephrasing the claim about scaling to complex codebases and making the scope explicit, and for the other clarifications! However, my main concern that the current technique has not been demonstrated on large-scale refactorings remains, and I would recommend the authors to conduct such experiments. The stated cost of individual refactoring (100 dollars) for refactoring of ~10 files is significant. The authors should investigate how the refactoring quality changes with cost. I'm maintaining my score.

---

> > > ### Author Response · Authors · 2025-11-28
> > >
> > > Thank you for the follow-up and for acknowledging the significance of the cost.
> > >
> > > Regarding the request to study how refactoring quality varies with cost: the paper already contains this analysis in Figure 3, where performance is plotted as a function of the number of sampled refactorings $K$. Since cost is determined entirely by the token usage for each $K$, adding dollar values simply makes that cost–quality curve explicit. Using token-usage reported by Claude Code (which internally runs Opus 4.1), the approximate costs for refactoring 10 Transformers files are:
> > >
> > > | $K$  |  Mean Cost (USD) |
> > > |---|---|
> > > |1|   $8.46|
> > > |2|   $16.90|
> > > |5|   $42.26|
> > > |7|   $59.18|
> > > |10|  $84.54|
> > > |14|  $118.34|
> > >
> > >
> > > We will add these values to the paper so that Figure 3 can be directly interpreted as a cost–quality curve: MDL improves with increasing cost, with diminishing returns.
> > >
> > > On the question of scale: refactoring these multi-function files from the Transformers and Diffusers libraries is clearly a real-world task, and represents a substantial step beyond prior library-learning work, which operated on 10–20-line synthetic programs. While we agree that full-repository refactorings remain an open challenge, our experiments already push near current model capabilities in both cost and performance.

---

### Official Review · Reviewer_NZ2R · 2025-11-01

**Soundness:** 2
**Presentation:** 3
**Contribution:** 2
**Rating:** 4
**Confidence:** 3

**Summary:**

This paper presents LIBRARIAN, a method for refactoring multiple code files into reusable libraries, and MINICODE, a benchmark for evaluating library design. The authors investigate what makes good refactoring metric through human study and asymptotic analysis, finding that MDL aligns best with developer preferences. They demonstrate their approach on competition programming and real-world repositories like HuggingFace Transformers and Diffusers.

**Strengths:**

- This paper is well written and achieves impressive real-world validation by refactoring HuggingFace production code with 67% MDL reduction while maintaining correctness.
- This work provides systematic comparison of multiple metrics through asymptotic analysis and human studies, finding MDL superior to traditional software engineering metrics.

**Weaknesses:**

- This evaluation covers only 10 Transformers files and 2 Diffusers tasks, which seems insufficient to support claims about general applicability to real software projects.
- This human study with only 12 participants lacks statistical power to distinguish between MDL and tokens metrics, yet the authors make strong claims about MDL superiority.

**Questions:**

- Could you provide concrete examples showing these improve code quality beyond just MDL score?
- How does LIBRARIAN's performance degrade with larger cluster sizes S? The paper fixes S=3 for CodeContests but S=5 for Transformers without justification.

---

> ### Author Response · Authors · 2025-11-21
>
> We thank the reviewer for their valuable time and feedback. We have addressed each point raised below and hope our clarifications resolve any concerns.
>
> > evaluation covers only 10 Transformers files and 2 Diffusers tasks… insufficient to support claims about general applicability
>
> Librarian refactors 10 Transformers and 11 Diffusers files (total 12,915 lines of code). But our initial submission overstated the extent to which LIBRARIAN is ready for deployment on full repos. We simply hoped to say that our work is head-and-shoulders closer to real-world use relative to past library learning algorithms, which worked with 10-20 lines of code, and almost never human written programs. Additionally, cost is a limitation in applying these methods.
>
> We’ve revised to say the following contributions: LIBRARIAN “outperforms prior library learners on existing benchmarks and scales to real-world human-written Python code” (previously “scales to complex codebases”) and gives “refactorings of real-world Python sources… successfully refactors files from the Huggingface Transformers and Diffusers libraries” (previously “refactorings of real-world codebases… successfully refactors the Transformers and Diffusers libraries”). We still say that “to the best of our knowledge this is the first time library learning has been successfully applied to real-world software projects” – because indeed this is true!
>
> The main limiting factor in scale is cost. We estimate the cost of a single experiment for refactoring 1 Transformer task (5 files) with $k=15$ samples to cost around 100 dollars (using Claude Code).
>
>
> > human study with only 12 participants lacks statistical power to distinguish between MDL and tokens metrics, yet… strong claims about MDL superiority
>
> The study had 14 participants and showed MDL better matches human preference than MI, but that MI is statistically indistinguishable from tokens (Fig 4). Fig 2 and 5 give analyses independent of the human study supporting MDL. Our finding is not that tokens are bad, but that MDL is at least as good.
>
> > could you provide concrete examples showing these improve code quality beyond just MDL score?
>
> In Diffusers, the lowest-MDL refactoring is better because it removes the repetitive wrapper code every scheduler must re-implement in the highest-MDL (worst case) design. It uses mixins to let schedulers inherit shared noise and timestep utilities directly, eliminating the duplicated argument-passing that clutters the function-based version. In Transformers, the lowest-MDL refactoring similarly centralizes shared logic—attention, MLP, rotary embedding, decoder-layer forward, and model-forward—into base classes, removing roughly 1,675 lines of duplicated code across the 5 models. The high-MDL variant instead repeats each 70-line attention block, 50-line decoder layer, 35-line rotary routine, and ~100-line model forward in every model.
>
> > does LIBRARIAN’s performance degrade with larger cluster sizes S? …
>
> As $S\rightarrow \infty$ the backend LLM cannot process the context and gives no/broken output. To save cost, we fix $S=5$, chosen heuristically. (In a few experiments we fix $S=3$ because that is what Regal used, for fairer Regal comparison.) On Huggingface files we tested $S=7,10$ but the Opus 4.1 in Claude Code consistently failed to refactor successfully.

---

### Meta-Review · Area_Chair_Ja6i · 2026-01-07

**Summary:**

This paper introduces LIBRARIAN, a method for refactoring multiple source files into reusable libraries, and MINICODE, an open-ended benchmark for evaluating library design quality. Reviewers broadly agree the problem is important. The main concerns focus on limited evaluation scale, small human-study size, cost and scalability, and whether claims about real-world applicability and MDL superiority are overstated. The rebuttal substantially clarifies scope, tones down claims, adds concrete qualitative examples, cross-model MDL analysis, cost–quality curves, and detailed responses to methodological questions, though fundamental scale limitations remain.

**Reviewer Concerns:**

Reviewer NZ2R’s concerns about limited real-world evaluation scale, small human-study size, lack of concrete quality examples, and unclear behavior with larger cluster sizes were largely addressed through claim revision, qualitative refactoring examples, clarified participant counts, and explicit discussion of cost and LLM context limits, with scalability remaining a constraint.

Reviewer vBJz’s concerns about limited scale, incremental novelty over Regal, clustering quality, single-use abstractions, missing citations, and model dependence were mostly addressed through clarified claims, added citations, clustering explanations, GPT-model discussion, and cost–quality analysis, though the lack of truly large-scale refactoring experiments remains unresolved, as explicitly noted by the reviewer.

Reviewer aB2G’s concerns about MDL’s dependence on a single model, semantic equivalence risks, missing citations, and metric definitions were addressed with new cross-model correlation results, explicit discussion of semantic risks, added citations, and clearer metric explanations, leaving only inherent limitations of test-based validation.

Reviewer p9FN’s concerns about coarse file-level clustering and clarity of the Transformers case study were addressed by acknowledging clustering limits in the paper and adding a detailed end-to-end walkthrough of the example, resolving their issues.

**Reviewer Scores:**

Reviewer NZ2R (4) did not state a score update, but since most technical questions were addressed and claims were revised, remaining concerns relate mainly to acknowledged scalability limits. (BUT IMPORTANT!)

Reviewer vBJz (4) explicitly stated they were maintaining their score, noting that despite clarifications, the lack of large-scale refactoring experiments remains unresolved.

Reviewer aB2G (4) did not indicate a score change, but since their methodological and citation-related concerns were directly addressed, most issues appear resolved.

Reviewer p9FN (10) did not state a score change and remained strongly positive, with their minor concerns fully addressed in the rebuttal. (I DIDN"T CONSIDER THIS PAPER AS A FULL MARKING PAPER)

---

### Decision · Program_Chairs · 2026-01-26

Reject